

# Observational analysis of the daily cycle of the planetary boundary layer in the central amazon during a typical year and under the influence of the ENSO (GoAmazon project 2014/5)

Rayonil G. Carneiro[1], Gilberto Fisch[2]

[1] National Institute for Space Research (INPE), Sao José dos Campos, Brazil
[2] Department of Aerospace Science and Technology, São José dos Campos, Brazil

*Correspondence to*: Rayonil G. Carneiro (rayonilcarneiro@gmail.com)

**Abstract** The Amazon biome contains more than half of the remaining tropical forests of the planet and has a strong impact on aspects of meteorology such as the Planetary Boundary Layer (PBL). In this context, the objective of this study was to conduct observational evaluations of the daily cycle of the height of the PLB during its stable (night) and convective (day) phases from data that were measured and/or estimated using instruments such as a radiosonde, SODAR, ceilometer, wind profiler, Lidar and microwave radiometer installed in the central Amazon during 2014 (considered a typical year) and 2015 during which an intense El Niño Southern Oscillation (ENSO) event predominated during the GOAmazon experiment. The results from the four intense observation periods (IOPs) show that during the day and night periods, independent of dry or rainy seasons, the ceilometer is the instrument that best describes the depth of the PBL when compared with *in situ* radiosonde measurements. Additionally, during the dry season in 2015, the ENSO substantially influenced the growth phase of the PBL, with a 15% increase in the rate compared to the same period in 2014.

## 1. Introduction

The Amazon basin covers about a third of the South American continent and extends for approximately $6.9 \times 10^6$ km$^2$, of which about 80% are covered by tropical forests (Tanaka et al., 2014; Ghate and Kollias, 2016). The Amazon biome represents more than half of the world's remaining tropical forests, and consequently has a strong impact on the climate of South America, thus is one of the major tropical convective regions in the global climate system (Tang et al., 2016). It provides moisture to the global hydrological cycle and energy to drive the global atmospheric circulation, with with a large influence on meteorological components such as the Planetary Boundary Layer (PBL). Understanding convective systems over the Amazon region through observations is important for understanding and simulating this systems.

In this region there is a substantial quantity of convective activity that occurs during the entire year, but there are significant seasonal differences due to annual variation of atmospheric circulation and thermodynamic structure (Marengo and Espinoza, 2016), and these wet (or rainy) and dry season are well-defined. In this context, during the years 2014 and 2015, in the central Amazon region, the *Green Ocean Amazon* (GOAmazon) project was conducted with the objective of observing the influence of the complex interaction between the pollution plume generated in the city of Manaus-Amazonas, and clouds and vegetation (Martin et al., 2016). This project approached research questions from a multidisciplinary perspective, and one of the studied topics was the physics involved in convective processes in the Amazon with emphasis on differences between wet and dry seasons.

The PBL is a turbulent layer of the atmosphere near the surface that results from the interaction between the surface and the atmosphere. The knowledge of the properties of the PBL has important scientific and practical applications because through



this understanding of the PBL operational models of weather and climate forecasting can be refined, pollutant dispersion
processes can be adequately described, the eolic potential of a region can be objectively determined, patterns of ventilation in
urban areas can be estimated, and improvements in agricultural techniques can be made (Englberger and Dörnbrack, 2017).
Furthermore, a more realistic representation of processes that occur in the PBL can benefit numeric models of weather
forecasting with better parameterization of convection, clouds, and rain (Holtslag et al., 2013). The PBL characteristics,
related to surface processes, provide important information regarding the priming of the atmosphere for convective initiation
(Tawfik and Dirmeyer, 2014).
Holtslag et al. (2013) state that the PBL, since it is the lowest level of the atmosphere, is in continuous interaction with the
Earth's surface, with significant turbulent transfer of heat, mass, and *momentum*. According to these authors the PBL
presents, during its daily cycle, large variations in temperature, wind, and other variables in response to atmospheric
turbulence and convective processes that occur in a tridimensional and chaotic form in timescales that range from seconds to
hours, and the length across which these events occur are between a few millimeters up to the entire depth of the PBL (1-2
km) or more in the case of convective clouds.
Neves and Fisch (2015) emphasize that an important characteristic of the PBL is the determination of its height because it is
then possible to estimate the volume into which the source of pollution will be dispersed, and this is an important parameter
for modeling of atmospheric dispersion. During its daily cycle the PBL undergoes atmospheric processes generated by
thermal and mechanical convection during the day, and displays stable conditions at night. The height of the PBL during the
atmospheric instability phase (principally during the day) is denominated the Convective Boundary Layer (CBL), and that
during the stable period (principally at night) is named the Nocturnal Boundary Layer (NBL).
In this context, observational evaluation of the daily cycle of the PBL, with emphasis on the CBL and NBL, represents an
important field of study since within the PBL there occur processes that have large impact on society and the terrestrial
environment. Therefore, the objective of this study was to contribute to a more thorough understanding of the daily cycle of
the height of the PBL through integration and comparison of data that were measured and/or estimated using instruments
such as a radiosonde, SODAR, ceilometer, wind profiler, Lidar and microwave radiometer installed by the GOAmazon
experiment. Furthermore, this study attempted to verify if the observational methods were representative of the daily
variation in the cycle of the CBL and NBL in the central Amazon during 2014 (stated as a normal year) and 2015, and to
elucidate the influence of an intense El Niño Southern Oscillation (ENSO) event that occurred in 2015/2016.

## 2. Material and Methods

In order to conduct this observational study, data from the GOAmazon project 2014/5 were used. The article by Martin et al.
(2016) describes the details of the experiment wherein these data were collected, its principal objectives, and some results.
These data were collected using the structure that was installed at a research station called T3 (03º 12' 36" S; 60º 36' 00"
W), located north of the municipality of Manacapuru in the State of Amazonas, about 9.5 km from the urban area, and about
11.5 km from the left bank of the Solimões River, at the confluence of the mouth of the Manacapuru River (Figure 1) in the
central region of the Amazon basin.

At the experimental site of the T3 station, instruments were installed to obtain measurements of the hydrological cycle, PBL
energy flux, and other micrometeorological variables, and the data from these measurements are available on the site of
ARM - Climate Research Facility (https://www.arm.gov). For the current study, four intense observation periods (IOP) were
defined in order to capture the peak of the rainy (February and March) and dry (September and October) seasons of 2014 and
2015. The dates for each IOP were IOP1 and IOP3, February 15 to March 31, 2014 and 2015, and IOP2 and IOP4, from





September 1 to October 15, 2014 and 2015. For the daily cycle analysis of the PBL it was considered the sunrise at 06 LT
and the sunset at 18 LT, not varying throughout the year.
In order to measure the height of the PBL, remote sensing estimation methods including a Wind Profiler (WP), Ceilometer,
SODAR, Microwave Radiometer Profiler (MWP) and Lidar were used, and these data were compared to data taken *in situ*
obtained using radiosondes (RS), and this method is described below.

### 2.1 Radiosonde (*in situ*) –RS

In this experiment RS measurements were obtained using a system that included a DIGICORA (MW12) (Vaisala Inc.,
Finland) with radiosonde model RS92SVG. The RS were coupled to a meteorological balloon that had an average ascension
rate of 5 m s$^{-1}$, and the readings were taken at 02, 08, 14 and 20 Local Time (LT), and during the IOPs 1 and 2 and extra RS
was done at 11 LT to better characterize the convective phase. From the RS measurements other data were taken which
included pressure, altitude, geographic position (latitude and longitude), air temperature (dry bulb and dew point), relative
humidity, and wind velocity and direction. From these data the potential temperature ($\theta$) and specific humidity (q) were
extrapolated, and using the vertical profiles of $\theta$ and q the height of the PBL was calculated by identifying the vertical level
where there was a systematic increase in potential temperature and a sudden reduction in specific humidity, and this method
is called the two profile method, as described in detail by Santos and Fisch, (2007), Seidel et al., (2010) and Wang et al.

94    (2016).

### 2.2 Wind Profiler (WP)

In order to construct a wind profile (WP) at the study site, a Radio Acoustic Sound System (RASS) model RWP915 from
Vaisala Inc. (Finland) was used for direct and continuous measurements of the PBL. The WP are Doppler instruments used
to detect the vertical wind profile, and they function at a frequency of 50 MHz to 16 GHz. The WP/RASS installed at the
study site operates at 915 MHz to measure the wind profile. The RASS transmitter aids in the measurement of the profiles of
the vertical temperature. The RWP/RASS operates through transmission of electromagnetic waves in the atmosphere and
measures the intensity and frequency of the backscatter of the waves, assuming that atmospheric dispersion elements are
moving with the average wind profile.
Since this is an instrument that operates at a high frequency and using smaller intervals of space between layers, it is
frequently used for tropospheric observations, especially for the PBL. The method used in this study is described by Wang et
al. (2016), wherein the height of the PBL was estimated using the vertical profile of electromagnetic refraction of RWP,
where the maximum of this index occurs in the upper part of the PBL.

### 2.3 SODAR

In the study area a Mini SODAR (Sound Detection and Ranging) (SCINTEC, Germany) was installed. This monostatic
equipment consists of an emission/receiving antenna with an area of 1.96 m$^2$, and functions at a power of 10 W and a
frequency of approximately 2 kHz. Using the SODAR profiles of wind velocity and direction were obtained at intervals of
30 min at a maximum height of 400 m.
Through remote sensor measurement by the SODAR the height of the PBL was calculated for its night phase (NBL) through
the determination of the maximum wind height (jet). This method was suggested by Neves and Fisch (2011) and showed
good results for the Amazon due to its operational limit (400 m) and taking into account that the NBL in the region has an
average depth of 100 to 300 m.

### 2.4 Ceilometer





The PBL was also monitored using a Ceilometer model CL31 from Vaisala Inc. (Finland). Ceilometers are a LIDAR remote
sensing instruments that register the intensity of optical backscattering at the near infrared wavelength through an emission
of a vertical pulse that is autonomously executed. The data generated yield the height of the cloud base, and using this data
the height of the PBL is estimated (Shukla et al., 2014; Carneiro et al., 2016). The ceilometer is a high-frequency instrument
with a sampling rate of 16 s and is a powerful tool for measuring the height of the PBL during its daily cycle (day and night
phases) to a high level of detail. The intensity of backscattering depends on the concentration of particles in the air, but also
depends on the reflexive properties of the atmosphere, which are related to it level of humidity. Therefore, this instrument is
useful for creating a tridimensional map of aerosols, air pollutants, and industrial and natural emissions of particles.
Ceilometers use a retrodiffusion laser to determine the coefficient of the attenuated portion, and the coefficients for aerosols
are obtained from these data, and subsequently the heights of the cloud base and the PBL are calculated.
**2.5 Microwave Radiometer Profiler (MWR)**
Data were also used from a Microwave Radiometer Profiler model MP3000A from Radiometrics Corp., Boulder, CO, USA.
This instrument provides vertical profiles of temperature, humidity, and liquid water content at a sampling rate of 60 s and
average values at intervals of approximately 5 minutes.
The profiles are deduced from measurements of radiance values of absolute microwaves (expressed as "brightness
temperature") obtained at 12 different frequencies at intervals of 22-30 and 51-59 GHz. This type of data is useful as input
into numerical models of weather forecasting which need high resolution profiles in continuous time. The data for the air
temperature profile from the MWR were combined with the vertical pressure profile from the RS sampling in order to
interpolate these to the entire daily cycle, this allowing for the calculation of the vertical field of temperature potential and
obtain the PBL height through the profile method.
**2.6 Lidar**
LIDAR was also used to estimate the height of the PBL using a LIDAR from Halo Photonics (United Kingdom), a single
autonomous instrument from the most recent line of products from this company for atmospheric remote sensing. These
systems are adequate for meteorological studies of the PBL and also for measurements of cloud cover, vertical wind profiles,
and air quality monitoring (Gouveia et al., 2017).
Lidar is an active remote sensing instrument that provides measurements of radial velocity and attenuated backscattering in
real time. The fundamentals of its operation are similar to those of radar in which pulses of energy are transmitted to the
atmosphere, the energy that is bounced back to the receiver is collected and measured as a resolved signal in time. Starting
with the interval of time between each exiting pulse and the retro-diffused signal, the distance to the disperser is inferred.
The radial velocity or of the line-of-sight of the dispersers is determined using the movement of the Doppler frequency of the
retro-diffused radiation. LIDAR uses a technique of heterodyne detection (method of extraction of coded information as a
phase modulation and/or the frequency of a wavelength) in which the return signal is mixed with a reference laser beam (a
local oscillator) of a known frequency. A computer within the instrument then processes the signal determining the Doppler
frequency change using the spectrum from the signal. The energy content of the Doppler spectrums can also be used to
determine attenuated backscattering.
Lidar operates in the near infrared wavelength and is sensitive to retro-diffusion of aerosols at the micrometer scale,
therefore it is capable of measuring wind speeds under clear sky conditions with a very high precision (normally 10 cm s$^{-1}$).
Lidar also possesses a superior capacity for hemispheric sweeping, thus permitting tridimensional mapping of turbulent
fluxes within the PBL. Using the variance of the vertical wind speed ($\sigma w^2$) provided by the Lidar, the method of Huang et al.
(2017) was employed, where the authors define the depth of the PBL as a layer in which $\sigma w^2$ exceeds a specific limit (0.1 m$^2$
s$^{-2}$).



Table 1 presents a synthesis of the instruments used in this study, the observation periods, and temporal and spatial
resolutions.
Determination of monthly rainfall for both study years was done using data taken with a disdrometer model Parsivel[2] (OTT
Hydromet GmbH, Germany), with a temporal resolution of 10 minutes. The eddy covariance system provides an estimate of
the net exchange of energy and mass between the terrestrial surface and the atmosphere. The estimated flux is given by a
scalar magnitude that is defined as the average of the product of the fluctuations of the vertical velocity and the concentration
that is being transported. In practice, this technique consists of taking observations of variables of the product at a high
frequency, and from this large number of samples of each variable the statistical covariation is calculated between them. In
this manner the system provides *in situ* measurements of turbulent fluxes of *momentum,* sensible and latent heat, and carbon
dioxide to the surface.
The measurements of radiation and soil heat flux were done every 30 minutes using the Surface Energy Balance System
(SEBS), which consists of measurements of solar and terrestrial radiation collected using radiometers, and the radiation
balance by a net radiometer. There was coupling of sensors measuring soil temperature and heat flux in this system.

**3. Results and Discussion**
Analysis of the meteorological variables revealed that accumulated precipitation was different between years (Figure 2). The
year 2014 was similar to the normal climatology (for the city of Manaus - data extracted from INMET, 2018) for the region
(2,300 mm), with a total of 2,451 mm. This high rate of rainfall can be understood as a response of the dynamic fluctuation
of the nearly permanent center of convection, associated with a high rate of local evapotranspiration, which contributed to
recycling of water vapor and rainfall (NobrE et al., 2009; Rocha et al., 2017). In contrast, the year 2015 registered a
significant reduction (approximately 30%) of the total rainfall in relation to the previous year, with a total accumulation of
1,764 mm, well below the normal climatological average. This reduction is associated with the occurrence of the El Niño
Southern Oscillation (ENSO) event of that year (ECMWF, 2017, Macedo and Fisch, 2018; Newman et al., 2018).
During 2014, monthly accumulated precipitation was always above 50 mm month$^{-1}$, and during representative months of
IOP1 (February and March, 2014) the total accumulated precipitation was 720 mm. However, during IOP2 (September and
October, 2014) the total accumulated precipitation was 185 mm, yielding a reduction of nearly 75% of accumulated
precipitation during IOP1. According to Ferreira et al. (2005), this difference between the rainy and dry seasons occurs
because rainfall distribution in the Amazon is very irregular, with high spatial and temporal variability. Marengo et al.
(2017) provide a more detailed explanation of this characteristic to central Amazonia and include the large-scale forcing role
in the description of rainfall during the rainy and dry seasons.
While 2015 had a large reduction in rainfall, during the months of the period of intense observation of the rainy season
(IOP3) the accumulated precipitation was 398 mm, representing a reduction of approximately 50% of the total accumulated
precipitation during the rainy season of 2014 (IOP1). During IOP4, the total accumulated precipitation was well below the
normal climatological value, as well as in comparison to the same period in 2014 (IOP2), with a total registered precipitation
of 68 mm, a reduction of approximately 65% compared to IOP2. This occurred due to the EN event being more intense
during these months (ECMWF, 2017; Newman et al., 2018). The ENSO event (2015/2016) was considered one of the most
intense of recent years, with an intensity similar to that which occurred during 1982/83 and 1997/98 (ECMWF, 2017).





### 3.1. Typical year (2014)

Daily cycles of 30-minute averages of the components of the balance of energy are presented in Figure 3 for the period IOP1 (Figure 3A) and IOP2 (Figure 3B) for 2014. In these figures, the radiation balance (Rn) had positive values between 06 and 18 LT, with 455W m$^{-2}$ for IOP1 at 12 LT, and IOP2 had greater intensity of Rn with 534.5 W m$^{-2}$ at 11 LT.

The latent heat flux (LE) showed that the majority of the available net radiation (daytime conditions) was used for this flux. Both periods had similar maximum values, with 355.9 (IOP1) and 350 W m$^{-2}$ (IOP2) at 12 LT. However, there was a reduction of net radiation converted into LE between the periods, with 75% in IOP1 and 66% in IOP2. This occurred because IOP2 had lower availability of soil moisture for evapotranspirative processes.

In the Amazon, especially during the rainy season, only a small fraction of Rn (about 10%) is transformed into sensible heat flux (H), and this maximum of 52 W m$^{-2}$ occurred at 10:30 LT during IOP1, while for IOP2, due to lower precipitation values and soil moisture deficit, there was an increase of 21% of the fraction of Rn transformed into H, with a maximum of 112.8 W m$^{-2}$ at 12 LT.

Nevertheless, only a small percentage of Rn is converted into heat flux in the soil (G), with a maximum of 50 W m$^{-2}$ for both IOPs. These results showed that independent of the season, this flux is always low, and is limited to about 5% of the total available energy.

Figure 4 shows the hourly average of the heights of the PBL for the IOP1 (Figure 4A), and IOP2 (Figure 4B). Diurnal (convective) and nocturnal (stable) conditions were separated by vertical lines at 06 and 18 LT, The RS (*in situ* measurements) was considered as truth depth of the Boundary Layer, while the others presented estimated by remote sensing.

During the phase in which the NBL is formed (between 00 and 06 LT), the IOP1 showed small vertical oscillations of its depth due to the occurrence of sporadic rainfall (Figure 4A). The results obtained from the ceilometer between 00 and 03 LT showed that the depth of the PBL varied between 180 and 280 m, and after 04 LT there was an increase in the maximum height to 350 m, which was reduced in the following hours to 275 m by 06 LT (sunrise).

The measurements made with the WP also showed some oscillations in the height of the NBL, with a reduction in height between 00 and 02 LT from 280 m to 250 m, and then an increase to a maximum of 350 m at 04 LT, remaining constant until 06 LT. The SODAR results during this interval showed lower variation of NBL depth during this same interval. The variation observed from the measurements by the different sensors is related to intermittent mechanical turbulence which could be the result of the presence of clouds and rain during some days and not on others, thus provoking an increase in wind variability during the night which has the effect of deepening the NBL. However, in this same interval during IOP2 (Figure 4B) the NBL was very stable with an average height of 250 m for all sensors (ceilometer, WP, SODAR, MWR and Lidar), thus corroborating the explanation of the influence of rainfall on the determination of variability of depth of the NBL.

The results found for the height of the NBL were similar to those reported by Neves and Fisch (2011), in a study using SODAR in southwestern Amazon, where the authors observed NBL heights varying from 150 to 329 m. However, Acevedo et al. (2004), also studying in a pasture site in the Amazon (in Santarém-PA), observed lower NBL heights than those from the current study (between 50 to 150 m), and this difference occurs because of different geographic conditions (Santarém suffers the influence of river breeze). An example of the influence of the geographic factor is the fact that, in this case, in the Santarém region the authors related several cases of formation of fog during the night, and this is one criteria that determines the height of the NBL, but this phenomenon did not occur at the pasture site in Rondônia nor at T3 site in the current study.



The phase of the erosion of the NBL begins at 06 LT when there is incidence of solar radiation after sunrise, but during IOP1
the most significant increase in Rn and consequently of H, occurred at 07 LT (Figure 3A). As a function of the small
increase in these fluxes during the first hours of the daytime period there is no increase in the PBL depth up to 08 LT,
indicating that the NBL that formed there was still not degraded. After 08 LT an increase in PBL height begins to be
registered, but only with an average rate of 22.8 m h$^{-1}$. This occurs due to a lower availability of energy in the IOP1, which
causes the erosion of the NBL to progress slowly, and total erosion occurs only at 09 LT when average rate is 102 m h$^{-1}$.
According to Stull (1988), the complete erosion of the NBL occurs when the PBL has a high growth rate (say above 100 m
h$^{-1}$).
In contrast, in the IOP2, as a function of greater stability of the NBL and the positive values of Rn and H occurring earlier,
initial erosion of the NBL begins at 06 LT and there is a rapid increase in the depth of the PBL compared to IOP1, and this
increase continues in the subsequent hours at a rapid rate of 70.8 m h$^{-1}$, which causes the NBL to be completely eroded by 08
LT. This result demonstrates that the erosion of the NBL in this region is conditioned by greater availability of energy in the
early hours of the morning, and by how much of this energy will be used for heating of the atmosphere (H).
In IOP1, after complete erosion of the NBL the development phase of the convective boundary layer (CBL) begins, and due
to the slow erosion of the NBL the growth of the CBL begins at 11 LT with a typical height of 850 m and an average
increase of 102 m h$^{-1}$, until it reaches its greatest depth at 1,180 m at 13 LT. However, soon after the maximum is registered,
the CBL presents a small reduction in depth (13.3 m h$^{-1}$), due to the low value of H, which did not exceed 50 W m$^{-2}$. This
surface flux, added to the entrainment flux at the top of CBL, was not sufficient to maintain turbulence in this layer, that
showed a reduction after this time.
During IOP2, with the NBL being rapidly degraded, the CBL that subsequently formed had a more rapid development, with
an average growth rate of 175.2 m h$^{-1}$. The CBL had a more prolonged phase, with maximum depth registered at 13:30 LT
of 1,590 m. After this maximum of the CBL there was a slight reduction in its growth rate (-39,2 m h$^{-1}$) until 17:30 LT,
when H returned to a null value. This depth is similar to that reported by Fisch et al. (2004), in a study of the CBL in a
pasture in Rondônia in the southwestern Amazon, where the authors observed a maximum depth in the dry season of 1,650
m. Neves and Fisch (2015) also observed in Rondônia that in the initial formation of the CBL during the dry season there
was very rapid growth between 08 and 11 LT, with maximum heights at 14 LT of about 1,500 m
**3.2. El Niño year (2015)**
The energy fluxes for IOPs 3 (Figure 5A) and IOP 4 (Figure 5B) show that during the 2015 rainy season, Rn IOP3 behaved
in an analogous manner as in the rainy season of 2014 (IOP1), with a maximum of 488 W m$^{-2}$ at 12 LT. However, during the
dry season of 2015 (IOP4) there was an increase in intensity compared to IOP2, with maximum Rn equal to 555.2 W m$^{-2}$ at
12 LT. This greater flux of net radiation to the surface will be converted into heat flux, meaning that there is greater energy
available during the dry season of 2015 compared to that of 2014 due to less cloud cover, a common characteristic in years
with ENSO events (Macedo and Fisch, 2018).

The LE registered a maximum of 310 W m$^{-2}$ during IOP3 at 12 LT. This result represented 65% of the partitioning of Rn.
These results are within the range of results for LE in this region, for which 70% of Rn is generally converted into LE (Von
Randow et al., 2004; Andrade et al., 2009). Due to the low the low water availability during IOP4 there was a reduction in
the partitioning of Rn into the LE flux, only about 35% in comparison with IOP3, resulting in a reduction of more than 50%.
The maximum LE registered was at 12 LT and was 179 W m$^{-2}$, while during IOP3 just 17% of Rn was converted into H,
with a maximum H observed at 13 LT of 86 W m$^{-2}$. During IOP4, H had larger averages and 60% of Rn was converted into





H, with a maximum of 280 W m$^{-2}$ at 12 LT. Furthermore, in 2015 during IOPs 3 and 4, only a small percentage of Rn was
converted into G, with a maximum of 40.0 W m$^{-2}$ in IOP3 and 50.0 W m$^{-2}$ in IOP4, both at 12 LT.
The daily cycle of CBL during IOP3 (Figure 6A), as well as in IOP1, showed vertical oscillations of the NBL´s heights
(between 00 and 06 LT), of 200 m (00:30 LT) to 375 m (03 LT). The WP and the SODAR yielded lower depths than did the
ceilometer and the MWR. However, during IOP4 (Figure 6B) the NBL was more stable, with an average height of 250 m,
similar to what was observed during IOP2. The result from 0 to 06 LT confirms that in the Amazon region the NBL is more
stable during the dry season compared to the rainy season, when it has larger variation in its depth.

The IOP3 demonstrated a similar erosion pattern for the NBL to that observed during IOP1, with the NBL still established
between 06 and 08 LT. From this time onward there was an increase in depth of the CBL with an average growth rate of 19.6
m h$^{-1}$. In this manner, just as in IOP1, the erosion of the NBL during IOP3 occurred slowly, with total degradation at 09 LT.
This result is similar to that which was observed during the same phase in IOP1, where the NBL was less stable, and together
with lower availability of energy at the surface, caused a slower erosion of the NBL. However, the erosion phase during
IOP4, in response to an increase in Rn and H, occurred earlier (06 LT). Additionally, during the dry season of 2015, the rate
of ascension was higher during the subsequent hours and reached 76.1 m h$^{-1}$, such that the NBL was completely eroded at 08
LT.
The development during IOP3 occurs in an analogous manner to that observed during IOP1, with weak vertical development
of the maximum depth. At 10 LT the CBL begins to develop and has a height of 830 m and a growth rate of 100.3 m h$^{-1}$,
reaching a maximum depth of 1,069 m (13:30 LT), demonstrating a shallower CBL during the wet season. However, in
contrast to IOP1, IOP3 had a CBL established during the entire evening period, probably as a function of lower frequency of
rainfall, and IOP4, during the 2015 dry season, had greater development of convection after erosion at 08 LT, with a high
growth rate of 193 m h$^{-1}$, and at 11 LT the CBL was completely established. With the CBL established as a result of greater
heating of the atmosphere by H, there was greater depth of the CBL, reaching a maximum of 1,925 m at 14 LT, and this was
influenced by the strong EN event, which intensified the dry season in the region, and has caused the highest values of H.
This caused greater thermal convection in the development of the CBL, which increased its depth by 21% in relation to that
observed in IOP2. This maximum depth of the CBL during IOP4 is not commonly found in studies conducted in the Amazon
region before, however Lyra et al. (2003), using radiosonde data, reported a maximum height of 2,200 m during the dry
season of 1994 in Rondônia.
**4. Conclusions**
During the four IOPS the results show that during daytime and nighttime intervals, independent of weather conditions, the
ceilometer is a promising sensor with good accuracy for direct and continuous measurement of the height of the PBL (which
on average ranged from 250 meters - NBL to 1,900 meters - CBL) when compared to *in situ* from RS. The RS, in spite of it
being a proven high-precision method, it did not capture the daytime evolution of the height of the PBL, due to the long time
interval between launches. While the MWR, WP and the Lidar were satisfactory for estimates of the convective phase (CBL)
of the PBL, during the nocturnal phase (NBL) these sensors overestimated heights. Additionally, the SODAR under- and
overestimated the NBL during these periods.
The intense EN event of 2015/2016 influenced the development phase of the CBL during the dry season of IOP4, and it had
a growth rate of about 15% higher than the results from IOP2, and a sensible heat flux (responsible for heating the air) that
was higher than the standard values for the central Amazon. As a consequence, more intense convective movements were
occurred and had contributed to a stronger vertical development of the layer.





The NBL erosion showed differences between seasons, presenting an erosion time of 2 hours in the dry IOPs (2 and 4), and 3
hours in the wet IOPs (1 and 3). A more detailed analysis of CLN erosion is being elaborated in Carneiro and Fisch (2019).

**Data availability**. The data sets used in this publication are available at the ARM Climate Research Facility database for the
GoAmazon2014/5 experiment (https://www.arm.gov/research/campaigns/amf2014goamazon, last access: 1 June 2019).

**Author contributions**. RGC and GF designed the numerical experiments and the first author performed the simulations as a
part of his PhD. RGC performed data analysis, assisted by GF. RGC and GF prepared the manuscript.

**Competing interests**. The authors declare that they have no conflict of interest.

**Acknowledgements.** Institutional support was provided by the National Institute of Space Research (INPE), the National
Institute of Amazonian Research (INPA), and Amazonas State University (UEA). Rayonil G. Carneiro acknowledges a
Brazilian National Council for Scientific and Technological Development (CNPq) graduate fellowship (140726/2017-9).
Rayonil G Carneiro and Gilberto Fisch thanks the GOAMAZON Project group for provide the data available for this study.

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





**Table 1: Instrument parameters.**

| | **RS** | **WP** | **SODAR** | *Ceilometer* | **MWR** | *Lidar* |
|---|---|---|---|---|---|---|
| **Observation period** | Jan/2014 to Dec/2015 | Jan/2014 to Dec/2015 | Feb/2014 to Dec/2015 | Jan/2014 to Dec/2015 | Oct/2014 to Dec/2015 | Jan/2014 to Dec/2015 |
| **Variables observed** | Pressure; Temperature; Humidity; Wind speed and direction | Wind (u, v, w); Wind direction | Horizontal wind velocity (u,v); Wind direction; Vertical wind velocity (w); | Height of base of clouds; PBL height; Vertical visibility; | Temperature; Relative humidity; | Variation of vertical velocity ($\sigma_w^2$); |
| **Vertical resolution (m)** | ~10 | 60 | 10 | -X- | ~100 | 30 |
| **Temporal resolution** | 4 to 5 times/day | 1 hour | 30 min | 16 s | 60 s | 10 min |





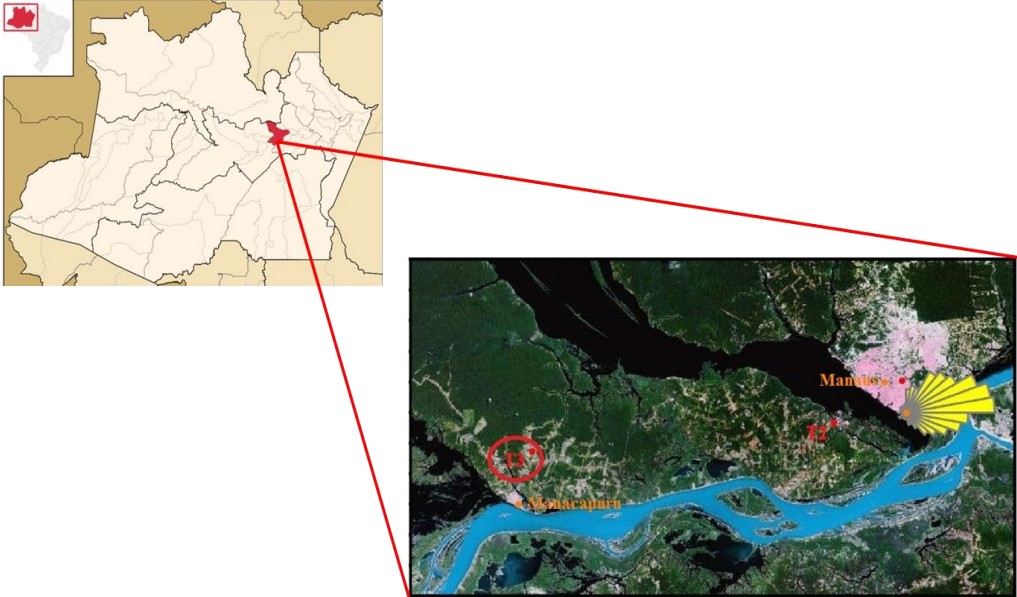

Figure 1: Location of the atmospheric measurement experiments in Manacapuru, Amazonas, Brazil.

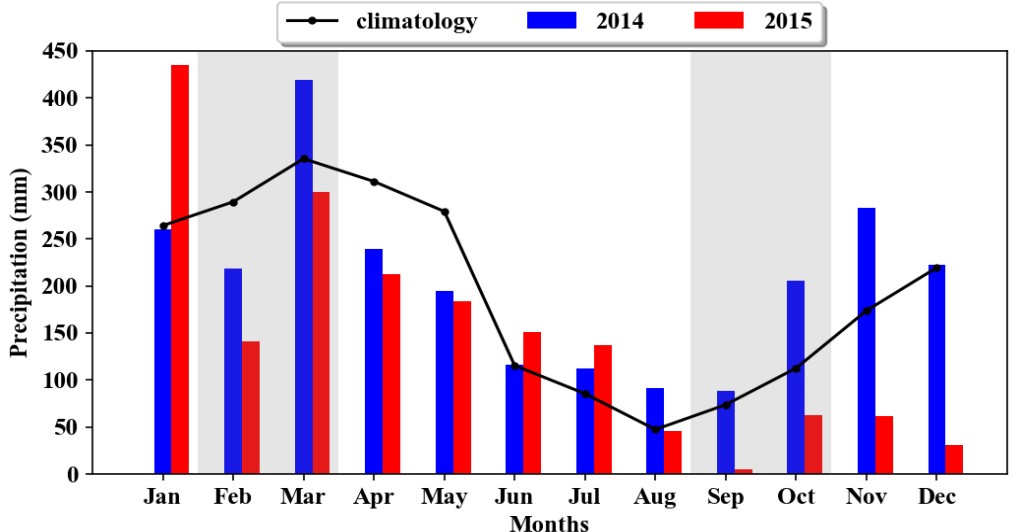

Figure 2: Distribution of accumulated monthly precipitation (mm) for the years 2014, 2015, and the normal climatological pattern.





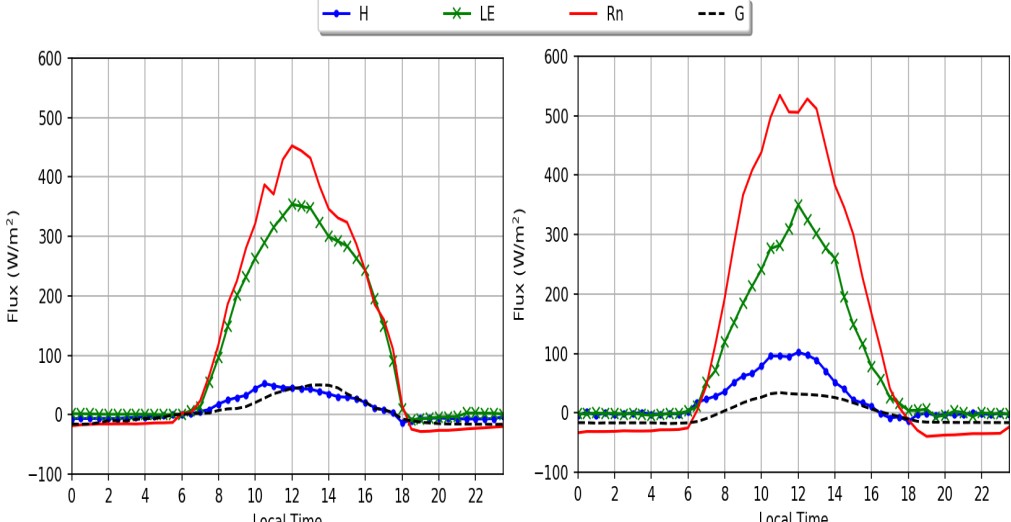

**Figure 3: Average daily cycle of the radiation balance (Rn) (W m$^{-2}$), sensible heat flux (H) (W m$^{-2}$), latent heat flux (LE) (W m$^{-2}$) and soil heat flux (G) (W m$^{-2}$) during the IOPs 1 and 2.**

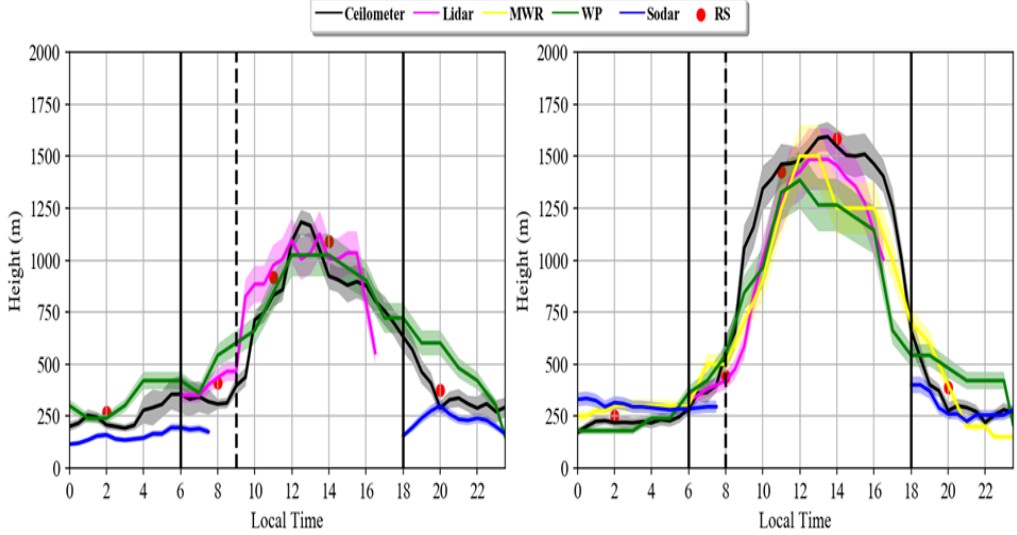

**Figure 4: Daily cycle of the height of the PBL during IOP1 (A) and IOP2 (B) experimental periods. The vertical lines represent sunrise (06 LT) and sunset (18 LT).**





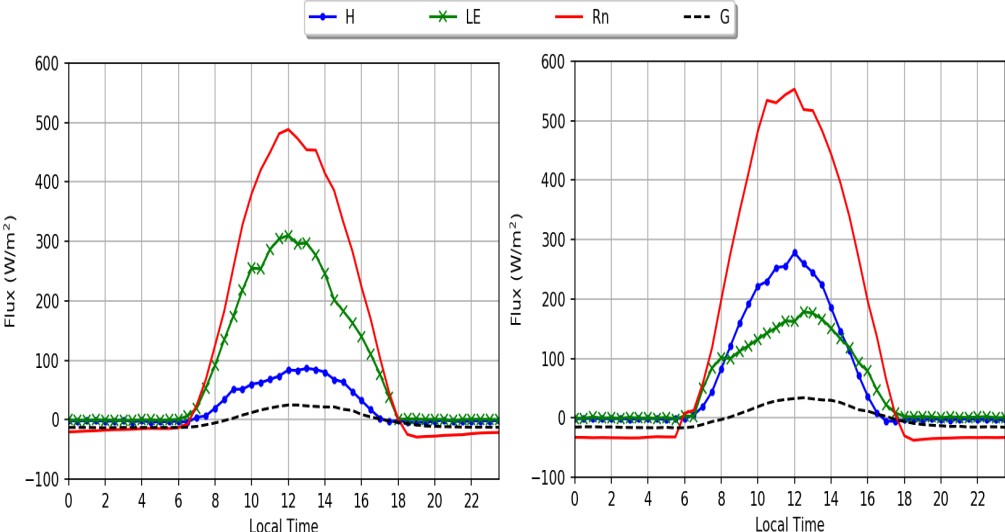

**Figure 5: Average of the daily cycle of net radiation (Rn) (W m$^{-2}$), sensible heat flux (H) (W m$^{-2}$), latent heat flux (LE) (W m$^{-2}$) and soil heat flux (G) (W m$^{-2}$) in the study region during the IOPs 3 and 4.**

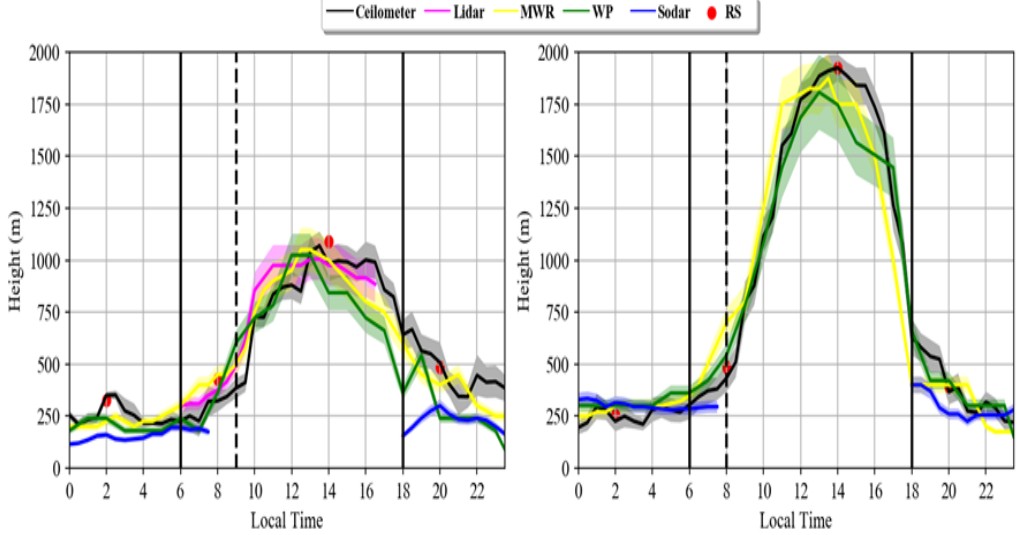

**Figure 6: Daily cycle of the height of the PBL during IOP3 (A) and IOP3 (B) experimental periods. The vertical lines represent sunrise (06 LT) and sunset (18 LT).**