# Peer review of "Observational analysis of the daily cycle of the planetary boundary layer in the central amazon during a typical year and under the influence of the ENSO (GoAmazon project 2014/5)"

_Atmospheric Chemistry and Physics, 2019_

## Referee Comment (RC1) · Anonymous Referee #1 · 10 Sep 2019

In this paper, the thickness of the PBL is observationally determined by 6 different methods in the Amazon region. Furthermore, the evaluation is performed over 4 different periods, comprising a substantially large time of observations. These reasons make this a very unique study, as very rarely such a comprehensive comparison is available. Therefore, the micrometeorological community might benefit from these results in terms of the evaluation it provides for different tools for PBL thickness determination. At the same time, there are many different communities that do research in Amazonia that could benefit from the knowledge of the typical daily evolution of this

quantity. The comparison between years with contrasting characteristics provides additional information in that regard. For these two reasons, I think this study should be published, as it might provide very important support to other researchers, in different fields.

I do have some suggestions, and they mostly relate to aspects that have not been shown and made me curious. Such a detailed dataset might answer many more questions than the authors raised, and my suggestions go in that direction:

1. The results are shown only in terms of typical daily cycles, which are important. However, it would be very nice to know the ranges of variabilities provided by each instrument. It could be done by adding error bars to the plots, but that would possibly make them "dirty". Another option would be to include additional plots of daily cycle of standard deviations for each platform used. This plot would tell the readers whether any of the platforms is more suscetible to errors that could just average out in the plots shown. Besides, that would give the readers an idea of the inherent variability observed for the PBL thickness in the region;

2. Along the same line of the previous comment, it would be very nice if the authors could go a step further and identify for the same years what drives deeper or shallower PBLs. This suggestion might be a bit more complex to address, so it may be done at a later study;

3. I would like to see case studies comparing the PBL thickness found by each method, for both a diurnal and a nocturnal event. This comparison would give the community a clear idea of what each method is capable of doing. I think this would be specially good for the nocturnal case, where there are still large uncertainties in the determination of the PBL thickness;

4. Is it possible to provide scatterplots comparing the thickness found by the radiosondes to those from other methods? In that case, I am assuming the radiosonde would be the "truth" and the comparison would be limited to the periods when radiosondes

are available. Nevertheless, given the long temporal coverage of the dataset, there may be enough points for this analysis which would, again, provide important insights on the quality of the PBL estimation provided by each platform.

---

## Short Comment (SC1) · 11 Sep 2019

Only today I became aware of this manuscript dealing with the retrieval of the MLH/PBL by means of ceilometer measurements. A large number of researchers has provided studies on diurnal and annual cycles of the MLH, on the potential of different measurement techniques, or on intercomparisons of methodologies (consequently more references should be given in the manuscript). To my knowledge a study on a statistically significant temporal development of the MLH in the Amazon region is indeed missing and I agree with the first anonymous reviewer that the study could be quite

useful. I also agree that a discussion of the accuracy of the retrievals and the variability of the MLH during the different IOPs should be included.

As I am (also) interested in the benefit of ceilometers for aerosol remote sensing including the derivation of the mixing layer height (MLH), I briefly want to comment on the manuscript. I restrict myself mainly to the "ceilometer part". Note, that below I have mainly mentioned my own papers. This has been done for the reason of simplicity (today is the last day of the ACPD-discussion) – of course there are many other papers useful to look at! An overview over potentially useful papers can be found, e.g., in the reference lists of my papers.

**Comments on the ceilometer part:**

- Sections 2.4 and 2.6: as far as the same methodology is used (exploitation of a backscatter signal) the explanations can be combined: a ceilometer is a backscatter lidar, i.e., they follow the same physical principles.

- line 120: what is meant with "high frequency instrument"? As this could be misleading I suggest to write that the "temporal resolution is high" or something similar.

- lines 121ff: "powerful tool ... high level of detail": This is quite a general statement that neglects the problems of retrieving the MLH, in particular when a low power ceilometer (CL31 compared to CL51 or Lufft ceilometers) is used. A brief overview of the inherent pitfalls should be given: Signal artefacts (Kotthaus et al., 2016, AMT), overlap problems [relevant in particular for the NBL] and water vapor absorption (Wiegner et al., 2019, AMT), or wrong attribution of detected layers (Geiß et al., 2017, AMT).

- Section 2.4: The description how the MLH is determined from the CL31 signals is missing. Is the proprietary software BL-VIEW used? How is it applied (compare Geiß et al., 2017)? A brief outline is strongly recommended as it help to understand the accuracy of the retrieval.

- line 123: What is meant by "reflexive properties"? Do the authors mean the refractive index? It could indeed be dependent on the relative humidity but I doubt that this effect has a significant influence on the MLH-retrieval. Or is the relative humidity mentioned because of potential water vapor absorption? Again, it is unlikely that this effect is relevant for the MLH-retrieval (Wiegner et al., 2014, AMT).

- line 124: "creating a tridimensional map": What does "tridimensional" mean? I assume that the ceilometer provides MLH as a function of time, or the particle backscatter coefficient as a function of time and height. Moreover, mentioning "aerosols, air pollutants, and industrial and natural emissions" might be misleading if it is interpreted as the potential to discriminate between different types of aerosol particles; this is impossible by a simple single-wavelength backscatter lidar (ceilometer).

- line 125: what is a "retrodiffusion" laser? Just skip this word. The expressions "coefficient of the attenuated portion" and "coefficients for aerosols" are not clear/known: do you mean "attenuated backscatter" or "particle backscatter coefficient"?

- line 126: "subsequently the heights of ... the PBL are calculated". See my previous comment. Please outline how this has been done.

- line 145: Similar to the ceilometer-section: What is a "retro-diffused signal"?

- Section 2.6: The authors mention "attenuated backscattering" (this means they use the lidar as an "elastic backscatter lidar") but do not use this quantity for the MLH-retrieval (why?)? Do I understand this correctly?

**General comments:**

- The authors should explain in detail how they determine the diurnal cycles of the MLH (for each instrument): which days are considered (only if full diurnal cycles can be determined)? Are the sample of days the same/similar for the different approaches? If not, is a bias expected? Did any instrument failures occur? What are the reasons for the gaps of some curves?

- The authors emphasize that the precipitation is quite high at the site. When it is raining the retrieval of the MLH by a ceilometer is not possible. How do these meteorological conditions influence the number of MLH-retrievals? How many full diurnal cycles could be determined during the IOPs? In this context the "sporadic rainfall" (e.g., line 220) could be discussed in more detail.

- The authors consider the MLH derived from radiosondes as truth. This is a frequently made assumption. However, the authors should explicitly mention that the methodologies (radiosonde vs. ceilometer/lidar) are based on different physical concepts.

- According to the shaded areas (not explained!) in Fig. 4 the variation/uncertainty (or whatever it indicates) is so large that the rapid growth/decrease of the MLH as discussed in lines 221ff is not significant.

- MLHs of $50 - 150$ m (line 234, Acevedo's results) can hardly be retrieved by a ceilometer (in particular when the overlap correction function is not very accurate). So it might be possible that the NBL is at 50 m at the authors' site but not detectable by the ceilometer. This fact is not covered by the discussion.

- Mentioning growth rates of e.g. 22.8 m/h (line 243) pretends an accuracy that is unrealistic. How is it determined: from the average over an IOP or from the mean of all individual diurnal cycles during the IOP (in the latter case the uncertainty

can be estimated)? How is such a "precision" justified in view of the vertical resolution of the ceilometer? Please explain.

- The limited temporal coverage of the lidar (compared to the ceilometer) retrievals should be explained.

- A comment on the availability of the different methodologies should be given: How many retrievals (obviously hourly averages) from the ceilometer, the lidar, and so on, are used for Figs. 4 and 6, (see also my first comment).

**Minor/technical comments:**

- Table 1: The vertical resolution of the ceilometer is given as "X". Please change.

- Figs. 4 and 6: Explain in the caption what the dashed line means? Instead of (A) and (B) one might also use "left" and "right".

- Fig. 6: panel B refers to IOP4 not IOP3.
* * *

---

## Referee Comment (RC2) · Anonymous Referee #2 · 17 Sep 2019

Observational analysis of the daily cycle of the planetary boundary layer in the central amazon during a typical year and under the influence of the ENSO (GoAmazon project 2014/5)

Authors: Rayonil G. Carneiro Gilberto Fisch

This study presents results from an intensive field campaign in Amazon (GO-Amazon project). It focuses on the detection and evolution of the Planetary Boundary Layer height (PBLH) during a non-El Nino (2014), and El Nino (2015) years. To estimate the

PBLH, this work used several remote sensing instrumentations (Vaisala wind profiler - WP, Scintec SODAR, Vaisala Ceilometer, Radiometrics micro-wave radiometer - MWR, and Halo wind-lidar - Lidar), and radiosondes (RS). Surface measurements, such as sensible (H) and latent heat (LE) fluxes from an eddy covariance system (EC), net radiation (Rn), and soil heat flux were also analyzed. This work is within the scope of the journal, since the realization of the importance of the PBLH in atmospheric chemistry. Unfortunately, I believe there are several concerns that the authors must address before publication.

Major Concerns:

1. I wonder what the meaning of a "typical year" is in the title. If ENSO correlates well with Amazon rainfall, then it is not an anomaly. I believe, the authors mean non-El Nino year (instead of typical year), and El Nino year (instead of ENSO).

2. The methods are incomplete:

a. Some instruments do not have the maker and/or model (SODAR, Lidar, EC, solar and terrestrial, soil heat flux).

b. How high are the radiation and flux measurements? What is the landscape of the study area, and what is the landscape composition of the flux and radiation footprint?

c. Is there any data filtering or all data were considered (clear, cloudy/rainy days)? What are the data sample size in figures 3 to 6?

d. Radiosonde PBL estimation, it only shows for the procedure for the convective boundary layer (CBL), what are the criteria for the nocturnal boundary layer (NBL)?

e. For the other remote sensing instrumentation, but SODAR. It seems that only the CBL detection are being shown. Do these methods also apply to the NBL?

f. Also, I believe that PBLH from MWR should be explained in more detail. Is this a novel method or is it based on previous studies? I wonder, because, from my experience, temperature profiles from the MWR do not show sharp gradients at the top of the CBL. Also, I am not sure how this interpolation works during the transition periods (mornings, and early night).

g. NBL detection from SODAR: The use of the maximum of the wind profile is associated with the lower level jet. Have the authors consistently observed LLJ at the study site? If so, this should be written.

h. Why not use the temperature profile from WP to estimate the NBL?

3. Results, discussion, conclusion: there are some parts that needs some work and/or clarification.

a. There are several comparisons among several parameter (precipitation, H, LE, PBLH) for wet/dry season or El Nino/non-El Nino years. However, averages are presented without any uncertainties, so the authors cannot affirm that those averages are different.

b. I do not understand the PBLH time series for the ceilometer. According to the methods, the PBLH is the cloud base, not the aerosol mixing layer estimated from the aerosol backscatter profile. Thus, are the black lines shown in figures 4 and 6 the cloud base diurnal variation? I do not believe that there will be boundary layer clouds during nighttime at such lower height (maybe fog – but not the cloud base). Also, there is a sharp drop during the afternoon, are the PBL clouds descending?

c. Are the CBL or residual layer (RL) shrinking for all remote sensing PBLHs during the afternoon? I wonder about this because H is still positive till about 16:30-17:00LT.

d. Lines 205-206: The authors claim that there is no ground water at the root level. Is it a shallow root or deep root vegetation? Depending of the type of vegetation, this might not be the reason for lower evapotranspiration.

e. Lines 208-211: Ground water stress are not be the only reason for low evapotranspiration.

f. Lines 223-230: So, how mechanical turbulence will affect the NBL for some instruments, and not for others? I believe a more in-depth analysis should be performed. If there is a presence of LLJ, then more likely there will not be too much turbulent mixing, and the NBL is going to grow due to radiative cooling. Also, usually the local maxima of the LLJ is not co-located with the bottom of the residual layer, so using the level of maxima wind speed will underestimate the NBL. Last, but not least, if there is rain, then most likely the remote sensing instruments do not work properly, and they will be unable to detect any mixing height.

g. The "erosion" of NBL discussion seems confusing and does not make any sense to me. Looking at figures 4 and 6, if the PBL is growing above 250m after 6am, then it means that there is no more a stable layer. If not, then how the PBL is growing? I guess the authors can check that looking at the SODAR RASS data...

h. Lines 254-255, 300-302: H by itself does not contributes for the PBLH growth, or the rate of change of temperature.

i. Lines 255-257: Not sure about that a small H will decrease the PBLH. According to the theory, if there is negative flux-divergence, the PBLH will increase. Also, how the entrainment at the top was calculated?

j. Lines 310-312: If one connects the red dots in figures 4,6, he/she can see a diurnal pattern in the RS PBLH time series.

Some Minor Concerns:

Lines 81-82: Wind Profiler, ceilometer, SODAR, MWP, and Lidar are not remote sensing estimation methods, but the instruments that probe the lower troposphere.

Table 1: Ceilometer vertical resolution is missing. I believe some of the instrumentation measure more than reported in row 3.

Figure 1: What is the location of the site? There are 2 red squares and red dots, are they the instrument location of Manacapuru site? If so, where are the location of the

[Figure]

instrumentation? If now, what are they?

Figures 3 and 5: What are the error bars for the measurements?

Figures 4 and 6: What is the shaded area? The errors? If so, how the errors are estimated? Why RS points do not have error bars?

---

## Author Comment (AC1) · 3 Dec 2019

Below are the responses to referee 1 (RC1)

1. The results are shown only in terms of typical daily cycles, which are important. However, it would be very nice to know the ranges of variabilities provided by each instrument. It could be done by adding error bars to the plots, but that would possibly make them "dirty". Another option would be to include additional plots of daily cycle of standard deviations for each platform used. This plot would tell the readers whether

[Figure]

any of the platforms is more suscetible to errors that could just average out in the plots shown. Besides, that would give the readers an idea of the inherent variability observed for the PBL thickness in the region;

I agree that inserting the error bars in the plots would make the figure "dirty", so the standard error was represented by shading (Fig. 4 and 6). However for the RS measurements the error bar was used as requested by RC2.

Therefore, I am sending you Appendix incluede two tables that show the standard deviation for 4 intervals of the daily cycle of PBL at each IOP. If you deem it necessary, I will add them in the final version.

2. Along the same line of the previous comment, it would be very nice if the authors could go a step further and identify for the same years what drives deeper or shallower PBLs. This suggestion might be a bit more complex to address, so it may be done at a later study;

In the present work, some forcing factors that influenced the deepening of PBL (Rn, H, etc.) were discussed. However, what suggested has been developed for a next study, through observations and modeling.

3. I would like to see case studies comparing the PBL thickness found by each method, for both a diurnal and a nocturnal event. This comparison would give the community a clear idea of what each method is capable of doing. I think this would be specially good for the nocturnal case, where there are still large uncertainties in the determination of the PBL thickness;

As previously answered (question 2), another article is being developed, which is analyzing typical days at each IOP during the transition between the night (NBL) and daytime (CBL) PBL periods.

4. Is it possible to provide scatterplots comparing the thickness found by the radiosondes to those from other methods? In that case, I am assuming the radiosonde wouldbe

the "truth" and the comparison would be limited to the periods when radiosondes are available. Nevertheless, given the long temporal coverage of the dataset, theremay be enough points for this analysis which would, again, provide important insightson the quality of the PBL estimation provided by each platform.

The scatter plot suggestion is quite valid, but the temporal coverage of IOP data is restricted to only 45 days and not a full year, thus making the sample with low representativeness.

Please also note the supplement to this comment:
https://www.atmos-chem-phys-discuss.net/acp-2019-578/acp-2019-578-AC1-supplement.pdf

**Supplement:**

Table 2 Standard deviation calculated for PBL height measurements of instruments at different intervals of the daily cycle.

| σ – IOP1 (February 15 to March 31, 2014) | | | | |
|---|---|---|---|---|
| Hours | 00 – 06 LT | 06 – 09 LT | 09 – 18 LT | 18 – 23 LT |
| Ceilometer | 27.2 | 18.0 | 10.7 | 9.7 |
| Lidar | -X- | 15.5 | 23.4 | -X- |
| MWR | -X- | -X- | -X- | -X- |
| RWP | 7.5 | 19.3 | 13.3 | 14.1 |
| Sodar | 2.5 | -X- | -X- | 4.0 |
| σ – IOP2 (September 1 to October 15, 2014) | | | | |
| Hours | 00 – 06 LT | 06 – 09 LT | 09 – 18 LT | 18 – 23 LT |
| Ceilometer | 5.2 | 21.5 | 18.4 | 11.4 |
| Lidar | -X- | 13.5 | 29.1 | -X- |
| MWR | 3.3 | 11.5 | 25.3 | 16.3 |
| RWP | 6.8 | 20.1 | 23.2 | 8.4 |
| Sodar | 2.6 | -X- | -X- | 5.8 |

* "X" represents where abscente measurements occurred

Table 3 Standard deviation calculated for PBL height measurements of instruments at different intervals of the daily cycle.

| σ – IOP3 (February 15 to March 31, 2015) | | | | |
|---|---|---|---|---|
| Hours | 00 – 06 LT | 06 – 09 LT | 09 – 18 LT | 18 – 23 LT |
| Ceilometer | 18.4 | 14.1 | 10.7 | 14.5 |
| Lidar | -X- | 11.1 | 24.3 | -X- |
| MWR | 4.3 | 6.8 | 14.7 | 7.8 |
| RWP | 2.4 | 17.2 | 18.4 | 11.6 |
| Sodar | 2.5 | -X- | -X- | 4.0 |
| σ – IOP4 (September 1 to October 15, 2015) | | | | |
| Hours | 00 – 06 LT | 06 – 09 LT | 09 – 18 LT | 18 – 23 LT |
| Ceilometer | 11.4 | 7.75 | 13.6 | 15.0 |
| Lidar | -X- | -X- | -X- | -X- |
| MWR | 2.5 | 16.4 | 27.7 | 12.6 |
| RWP | 3.1 | 8.9 | 29.6 | 7.8 |
| Sodar | 3.6 | -X- | -X- | 6.7 |

* "X" represents where abscente measurements occurred

---

## Author Comment (AC2) · 3 Dec 2019

Below are the answers to referee 2 (RC2)

2. Methods

a. Some instruments do not have the maker and/or model (SODAR, Lidar, EC, solar and terrestrial, soil heat flux).

Added in the document the makers and/or models

b. How high are the radiation and flux measurements? What is the landscape of the study area, and what is the landscape composition of the flux and radiation footprint?

Added in the document the height the radiation and flux measurements and landscape composition

c. Is there any data filtering or all data were considered (clear, cloudy/rainy days)? What are the data sample size in figures 3 to 6?

There was no filtering for cloudy / rainy days, since the instruments make measurements from surface to atmosphere, then cloudy days do not cause data changes. However, the data were verified and gap fillings was performed. The data sample size in figures 3 to 6 was 45 days, which corresponds to the intense observation period of the GoAmazon project (Lines 79-81).

d. Radiosonde PBL estimation, it only shows for the procedure for the convective boundary layer (CBL), what are the criteria for the nocturnal boundary layer (NBL)?

The Radiosonde PBL estimation showed CBL and NBL the criteria was used two profile method used for CBL and NBL by Santos and Fisch, (2007), Seidel et al., (2010) and Wang et al. (2016).

e. For the other remote sensing instrumentation, but SODAR. It seems that only the CBL detection are being shown. Do these methods also apply to the NBL?

The SODAR only was used for the NBL, because the vertical resolution of the instrument is 500 m.

f. Also, I believe that PBLH from MWR should be explained in more detail. Is this a novel method or is it based on previous studies? I wonder, because, from my experience, temperature profiles from the MWR do not show sharp gradients at the top of the CBL. Also, I am not sure how this interpolation works during the transition periods (mornings, and early night).

[Figure]

The method is based on previous studies.

g. NBL detection from SODAR: The use of the maximum of the wind profile is associated with the lower level jet. Have the authors consistently observed LLJ at the study site? If so, this should be written.

The height where the maximum wind value was found was considered the height of the NBL, according to Stull's definition, 1988 and applied in the Amazon by Neves and Fish (2015).

h. Why not use the temperature profile from WP to estimate the NBL?

Because our decision was to estimate the NBL and CBL by the same method in WP.

3. Results, discussion, conclusion: there are some parts that needs some work and/or clarification.

a. There are several comparisons among several parameter (precipitation, H, LE, PBLH) for wet/dry season or El Nino/non-El Nino years. However, averages are presented without any uncertainties, so the authors cannot affirm that those averages are different.

The averages are presented with uncertainties, where the shading in figures 3 to 6 indicates the standard error of the instruments. Which have been described in the captions of the respective figures.

b. I do not understand the PBLH time series for the ceilometer. According to the methods, the PBLH is the cloud base, not the aerosol mixing layer estimated from the aerosol backscatter profile. Thus, are the black lines shown in figures 4 and 6 the cloudbase diurnal variation? I do not believe that there will be boundary layer clouds during night time at such lower height (maybe fog – but not the cloud base). Also, there is a sharp drop during the afternoon, are the PBL clouds descending?

The Ceilometer PBLH time series does not refer to cloud base heights, but rather to

measurements of boundary layer heights. Thus the black lines shown in figures 4 and 6 are the daytime variation of the height of the PBL. Since the Ceilometer provides the height of the cloud base, the retrieval of the particle backscatters coefficient and PBL height (Wiegner et. Al., 2014; Shukla et al., 2014; Morris, 2016; Carneiro et al., 2016 Geiß et al., 2017). Thus, to improve the reader's understanding of the description has been improved methodology instrument (P4, L 118-129, text below).

"The PBL was also monitored using a Ceilometer model CL31 from Vaisala Inc. (Finland). Ceilometers are a type LIDAR remote sensing instruments that operating through a maximum vertical range of 7700 m and register the intensity of optical backscattering at the near infrared wavelength between 900 and 1100 nm through an emission of a vertical pulse that is autonomously executed. The Ceilometer provid data the height of the cloud base, the retrieval of the particle backscatter coefficient and PBL height (Wiegner et. al., 2014; Shukla et al., 2014; Morris, 2016; Carneiro et al., 2016; Geiß et al. 2017).

The ceilometer is a high-frequency instrument with a measurement interval of 2 s, and a sampling rate of 16 s and is a powerful tool for measuring the height of the PBL during its daily cycle (day and night phases) to a high level of detail. The intensity of backscattering depends on the concentration of particles in the air, but also depends on the reflexive properties of the atmosphere, which are related to it level of humidity. Therefore, this instrument is useful for creating a tridimensional map of aerosols, air pollutants, and industrial and natural emissions of particles. Ceilometers use a retrod-iffusion laser to determine the coefficient of the attenuated portion, and the coefficients for aerosols are obtained from these data, and subsequently the heights of the cloud base and the PBL are calculated."

c. Are the CBL or residual layer (RL) shrinking for all remote sensing PBLHs during the afternoon? I wonder about this because H is still positive till about 16:30-17:00LT.

The CLC measured by the instruments showed a decrease in its height around 16 LT,

where a reduction in H fluxes was also observed, but it is still positive. After 17 LT the H fluxes begins to become negative and the PBL begins its most stable phase (NBL).

d. Lines 205-206: The authors claim that there is no ground water at the root level. Is it a shallow root or deep root vegetation? Depending of the type of vegetation, this might not be the reason for lower evapotranspiration.

The T3 area was located was cover by shallow vegetation (pasture) (added in the description of the area in Material and Methods P2, L 72-73, text below).

"In order to conduct this observational study, data from the GOAmazon project 2014/5 were used. The article by Martin et al. (2016) describes the details of the experiment wherein these data were collected, its principal objectives, and some results. These data were collected using the structure that was installed at a research station called T3 (03° 12' 36" S; 60° 36' 00" W), located north of the municipality of Manacapuru in the State of Amazonas, about 9.5 km from the urban area, and about 11.5 km from the left bank of the Solimões River, at the confluence of the mouth of the Manacapuru River (Figure 1) in the central region of the Amazon basin. The T3 station is collocated in an area of pasture, surrounded by native forest with about 35 m of canopy height (Martin et al. 2016)."

e. Lines 208-211: Groundwater stress are not be the only reason for low evapotranspiration.

The Groundwater stress is not the only reason. Thus the text in question was improved (P6, L211-215, text below).

"The latent heat flux (LE) showed that the majority of the available net radiation (daytime conditions) was used for this flux. Both periods had similar maximum values, with 355.9 (IOP1) and 350 W m$-2$ (IOP2) at 12 LT. However, there was a reduction of net radiation converted into LE between the periods, with 75

f. Lines 223-230: So, how mechanical turbulence will affect the NBL for some instruments, and not for others? I believe a more in-depth analysis should be performed. Ifthere is a presence of LLJ, then more likely there will not be too much turbulent mixing,and the NBL is going to grow due to radiative cooling. Also, usually the local maximaof the LLJ is not co-located with the bottom of the residual layer, so using the levelof maxima wind speed will underestimate the NBL. Last, but not least, if there is rain,then most likely the remote sensing instruments do not work properly, and they will beunable to detect any mixing height.

Mechanical turbulence affected all instruments, however, each instrument has different measurement techniques, thus presenting differences in PBL height measurements at this range (00 - 60 LT). The study in question evaluated different PBLH measurement instruments, so the underestimation found for the NBL of the instruments that used the maximum wind speed level to estimate the height (SODAR, WP) showed that these sensors have limitations for measuring the NBL under mechanical turbulence.

g. The "erosion" of NBL discussion seems confusing and does not make any sense to me. Looking at figures 4 and 6, if the PBL is growing above 250m after 6am, thenit means that there is no more a stable layer. If not, then how the PBL is growing? Iguess the authors can check that looking at the SODAR RASS data: : :

According to Stull (1988), during the early morning the PBL is shallow. Its depth increases slowly at first because of the strong nocturnal stable layer that caps the young CBL. This first phase is sometimes referred to as the erosion of the NBL, wich begins after sunrise and the complete erosion occurs when the PBL has a growth rate above 100 m h$-1$.

The text in question has been changed for readers' better understanding (P7, L248-253, text below).

"The phase of the erosion of the NBL according to Stull (1988) begins after sunrise (at 06 LT in Amazon) and the complete erosion occurs when the PBL has a growth rate above 100 m h$-1$. In the IOP1 (Figure 4A), the erosion phase was occur 3 hours after

sunrise, where it was observed that in the first hours on this phase there is no increase in the PBL depth, only after 08 LT occured an increase in PBL heights (average rate of 22.8 m h$-1$). This occurs due to a lower H flux (Figure 3A), which causes the erosion of the NBL to progress slowly, and total erosion occurs only at 09 LT when average rate is 102 m h$-1$."

h. Lines 254-255, 300-302: H by itself does not contributes for the PBLH growth, orthe rate of change of temperature.

Positive sensitive heat flux (H) heats the air near the surface and by convection will rise, generating turbulent movements within the PBL which increases its depth. According to standard literature on studies of PBL (Stull, 1998; Holtslag, 1987; Foken, 2008), described below:

"During the daytime when H>0 the atmosphere is heated from below, while for H<0 the atmosphere near the surface is cooled (nighttime). This leads to unstable (daytime) and stable (nighttime) stratifications of the PBL. The magnitude and the diurnal cycle of H has important consequences on the height and the structure of PBL. During unstable conditions the air adjacent to the surface is heated and will rise. This rise continues as long as the air is warmer then the surroundings (up to an inversion layer). This process is know as convection, which is accompanied with the production of convective turbulence in the PBL."(Holtslag, 1987).

"The daily cycle is highly variable, after sunrise, the atmosphere is warmed by turbulance flux H from the ground surface, and the inversion layer formed during the night brakes up (erosion phase). The new layer is very turbulent, well mixed (mixed layer), and bounded above by an entrainment zone."(Foken, 2008; Stull, 1998).

i. Lines 255-257: Not sure about that a small H will decrease the PBLH. According tothe theory, if there is negative flux-divergence, the PBLH will increase. Also, how theentrainment at the top was calculated?

The decrease surface H reduce the PBLH as described in the above answer (h.). As the entrainment fluxes were not calculated, there is no way to indicate the values of this contribution. However, as answered by RC1, a next article is underway that will study the heating and cooling drivers of the layer through numerical modeling and observations.

j. Lines 310-312: If one connects the red dots in figures 4,6, he/she can see a diurnal pattern in the RS PBLH time series.

I agree, there is a daytime cycle pattern of the PBL, but it would not be correct to analyze the PBL variation by extrapolating the RS times, since the RS data range is long, and the PBL variation depends on many factors (H , Air temperature and etc.) which do not vary linearly and are highly variable throughout the day. So simply connecting the dots would not be representative of this cycle.

Some Minor Concerns:

Lines 81-82: Wind Profiler, ceilometer, SODAR, MWP, and Lidar are not remote sensing estimation methods, but the instruments that probe the lower troposphere.

Changed in the text

Table 1: Ceilometer vertical resolution is missing. I believe some of the instrumentationmeasure more than reported in row 3.

Changed in the Table 1

Figure 1: What is the location of the site? There are 2 red squares and red dots, arethey the instrument location of Manacapuru site? If so, where are the location of the instrumentation? If now, what are they?

Changed in the Figure 2

Figures 3 and 5: What are the error bars for the measurements? Changed in the Figure

Figures 4 and 6: What is the shaded area? The errors? If so, how the errors areestimated? Why RS points do not have error bars?

The shaded area is error. Put the error bars in RS

Please also note the supplement to this comment:
https://www.atmos-chem-phys-discuss.net/acp-2019-578/acp-2019-578-AC2-supplement.pdf

---

## Author Comment (AC4) · 4 Dec 2019

As requested by the referee following the changed figures.

[Figure]

**Fig. 1.** Figure 1: Location of the atmospheric measurement experiments in Manacapuru, Amazonas, Brazil.

[Figure]

**Fig. 2.** Figure 3: Average daily cycle of the radiation balance (Rn) (W m-2), sensible heat flux (H) (W m-2), latent heat flux (LE) (W m-2) and soil heat flux (G) (W m-2) during the IOPs 1 (A) and 2 (B). The s

[Figure]

**Fig. 3.** Figure 4: Daily cycle of the height of the PBL during IOP1 (A) and IOP2 (B) experimental periods. The vertical lines represent sunrise (06 LT), sunset (18 LT) (full line) and NBL erosion (dashed line)

[Figure]

**Fig. 4.** Figure 5: Average of the daily cycle of net radiation (Rn) (W m-2), sensible heat flux (H) (W m-2), latent heat flux (LE) (W m-2) and soil heat flux (G) (W m-2) in the study region during the IOPs 3 (

[Figure]

**Fig. 5.** Figure 6: Daily cycle of the height of the PBL during IOP3 (A) and IOP4 (B) experimental periods. The vertical lines represent sunrise (06 LT), sunset (18 LT) (full line) and NBL erosion (dashed line)

---

## Author Response (AR1)

Dear Mr ACP Editor

Firstly, I would like to thank the scientific reviews made, which contribute to a better article. Below I describe item by item, the questions of the Reviewers (RCs) and scientific community (SC). I am sending as a supplementary material a version of the article where the changes made are marked in the text.

**RC1**

*1. The results are shown only in terms of typical daily cycles, which are important. However, it would be very nice to know the ranges of variabilities provided by each instrument. It could be done by adding error bars to the plots, but that would possibly make them "dirty". Another option would be to include additional plots of daily cycle of standard deviations for each platform used. This plot would tell the readers whether any of the platforms is more susceptible to errors that could just average out in the plots shown. Besides, that would give the readers an idea of the inherent variability observed for the PBL thickness in the region;*

*I agree that inserting the error bars in the plots would make the figure "dirty", so the standard error was represented by shading (Fig. 4 and 6). However for the RS measurements the error bar was used as requested by RC2.*

*Therefore, I have also included two tables that show the standard deviation for 4 intervals of the daily cycle of PBL at each IOP. If you deem it necessary, I will add them in the final version.*

*2. Along the same line of the previous comment, it would be very nice if the authors could go a step further and identify for the same years what drives deeper or shallower PBLs. This suggestion might be a bit more complex to address, so it may be done at a later study;*

*In the present work, some forcing factors that influenced the deepening of PBL (Rn, H, etc.) were discussed. However, the reviewer suggestion is very reliable and , it will be included in a new study, comparing observations and modeling for a specific number of days (8 days for each IOPs).*

*3. I would like to see case studies comparing the PBL thickness found by each method, for both a diurnal and a nocturnal event. This comparison would give the community a clear idea of what each method is capable of doing. I think this would be specially good*

*for the nocturnal case, where there are still large uncertainties in the determination of the PBL thickness;*

*As previously answered (question 2), another article is under developing , which will analyze 8 typical days at each IOP during the transition between the night (NBL) and daytime (CBL) PBL periods.*

*4. Is it possible to provide scatterplots comparing the thickness found by the radiosondes to those from other methods? In that case, I am assuming the radiosonde would be the "truth" and the comparison would be limited to the periods when radiosondes are available. Nevertheless, given the long temporal coverage of the dataset, there may be enough points for this analysis which would, again, provide important insights on the quality of the PBL estimation provided by each platform.*

*The scatter plot suggestion is quite valid, but the temporal coverage of IOP data is restricted to only 45 days and not a full year, thus making the sample with low representativeness.*

**RC 2**

**1. I wonder what the meaning of a "typical year" is in the title. If ENSO correlates well with Amazon rainfall, then it is not an anomaly. I believe, the authors mean non-El Nino year (instead of typical year), and El Nino year (instead of ENSO). Yes, we have accepted this suggestion and we have changed the title of the paper to " ….for an El Nino and a non El Nino years"**

*2. Methods*

*a. Some instruments do not have the maker and/or model (SODAR, Lidar, EC, solar and terrestrial, soil heat flux)*

The manufactures and/or models for all instruments were added in the methodology section

*b. How high are the radiation and flux measurements? What is the landscape of the study area, and what is the landscape composition of the flux and radiation footprint?*

The height the radiation and flux measurements and landscape scenarios description were added in the new version of the document

*c. Is there any data filtering, or all data were considered (clear, cloudy/rainy days)? What are the data sample size in figures 3 to 6?*

We did not splitted (or filtered) the days between clear/cloud days. The goal was to present typical values for each season (that will include clear/cloudy day). Moreover, we have assumed that during the wet season, mostly of the days have rains/clouds, so they are cloudy. For the dry period, although there are small and isolated clouds, the situation represents clear conditions. The data sample size in figures 3 to 6 was 45 days, which corresponds to the intense observation period of the GoAmazon project (Lines 79-81).

*d. Radiosonde PBL estimation, it only shows for the procedure for the convective boundary layer (CBL), what are the criteria for the nocturnal boundary layer (NBL)?*

The Radiosonde PBL estimation showed CBL and NBL the criteria was used two profile method used for CBL and NBL by Santos and Fisch, (2007), Seidel et al., (2010) and Wang et al. (2016).

*e. For the other remote sensing instrumentation, but SODAR. It seems that only the CBL detection are being shown. Do these methods also apply to the NBL?*

The SODAR only was used for the NBL, because the vertical measurements were restricted only from the surface up to 500 m.

*f. Also, I believe that PBLH from MWR should be explained in more detail. Is this a novel method or is it based on previous studies? I wonder, because, from my experience, temperature profiles from the MWR do not show sharp gradients at the top of the CBL. Also, I am not sure how this interpolation works during the transition periods (mornings, and early night).*

The method is based on previous study, like Fisch (2012). Basically, it is a creation of a potential temperature profile using specific channels of MWR and then used the profile method to determine the height of the PBL.

FISCH, G. . The heights of the atmospheric boundary layer at a coastal region using remote sensing and in situ measurements. In: 16th Int Symp for the Advancements of Boundary Layer Remote Sensing, 2012, Boulder, USA. Proceedings of the 16th ISARS 2012. Boulder, USA: ISARS, 2012. p. 135-137.

*g. NBL detection from SODAR: The use of the maximum of the wind profile is associated with the lower level jet. Have the authors consistently observed LLJ at the study site? If so, this should be written.*

The height where the maximum wind value was observed was considered the height of the NBL, according to Stull's definition, 1988 and applied in the Amazon by previous study (Neves and Fisch,2015). Not all times this maximum windspeed can be considered as a low level jet, so this occurrence was not described in the document.

*h. Why not use the temperature profile from WP to estimate the NBL?*

Because our decision was to estimate the NBL and CBL by the same method in WP.

***3. Results, discussion, conclusion: there are some parts that needs some work and/or clarification.***

*a. There are several comparisons among several parameter (precipitation, H, LE, PBLH) for wet/dry season or El Nino/non-El Nino years. However, averages are presented without any uncertainties, so the authors cannot affirm that those averages are different.*

The averages are presented with uncertainties, as the shading in figures 3 to 6 indicates the standard error of the instruments. Significant tests were made with average/standard deviations between wet and dry periods. Which have been described in the captions of the respective figures.

*b. I do not understand the PBLH time series for the ceilometer. According to the methods, the PBLH is the cloud base, not the aerosol mixing layer estimated from the aerosol backscatter profile. Thus, are the black lines shown in figures 4 and 6 the cloudbase diurnal variation? I do not believe that there will be boundary layer clouds during night time at such lower height (maybe fog – but not the cloud base). Also, there is a sharp drop during the afternoon, are the PBL clouds descending?*

The Ceilometer PBLH time series does not refer to cloud base heights, but rather to measurements of boundary layer heights. During daytime (not nighttime and transitions time), the PBLH and the cloud base are coincidents. Thus the black lines shown in figures 4 and 6 are the daytime variation of the height of the PBL. Since the Ceilometer provides the height of the cloud base, the retrieval of the particle backscatters coefficient and PBL height (Wiegner et. Al., 2014; Shukla et al., 2014; Morris, 2016; Carneiro et al., 2016 Geiß et al., 2017). Thus, to improve the reader's understanding of this description, a new paragraph was included in the new version (P4, L 118-129, text below).

"The PBL was also monitored using a Ceilometer model CL31 from Vaisala Inc. (Finland). Ceilometers are a type LIDAR remote sensing instruments that operating through a maximum vertical range of 7700 m and register the intensity of optical backscattering at the near infrared wavelength between 900 and 1100 nm through an emission of a vertical pulse that is autonomously executed. The Ceilometer provide information about the height of the cloud base, the retrieval of the particle backscatter coefficient and PBL height (Wiegner et. al., 2014; Shukla et al., 2014; Morris, 2016; Carneiro et al., 2016; Geiß et al. 2017) for daytime conditions (not nighttime and/or transitions periods).

The ceilometer is a high-frequency instrument with a measurement interval of 2 s, and a sampling rate of 16 s and is a powerful tool for measuring the height of the PBL during its daytime to a high level of detail. The intensity of backscattering depends on the concentration of particles in the air, but also depends on the reflexive properties of the atmosphere, which are related to its level of humidity. Therefore, this instrument is useful for creating a tridimensional map of aerosols, air pollutants, and industrial and natural emissions of particles."

*c. Are the CBL or residual layer (RL) shrinking for all remote sensing PBLHs during the afternoon? I wonder about this because H is still positive till about 16:30-17:00LT.*

The CBL measured by the instruments showed a decrease in its height around 16 LT. This collapse is associated with the lower intensity of the thermal convection. Although there area reduction in H surface fluxes by this time, it is still positive (but weak). After 17 LT the H fluxes became negative and the PBL started its stable phase (NBL).

*d. Lines 205-206: The authors claim that there is no ground water at the root level. Is it a shallow root or deep root vegetation? Depending of the type of vegetation, this might not be the reason for lower evapotranspiration.*

The T3 area was located was cover by shallow vegetation (pasture) with shallow root depth. This info was added in the description of the area in Material and Methods P2, L 72-73, text below).

"In order to conduct this observational study, data from the GOAmazon project 2014/5 were used. The article by Martin et al. (2016) describes the details of the experiment wherein these data were collected, its principal objectives, and some results. These data were collected using the structure that was installed at a research station called T3 (03° 12' 36" S; 60° 36' 00" W), located north of the municipality of Manacapuru in the State of Amazonas, about 9.5 km from the urban area, and about 11.5 km from the left bank of the Solimões River, at the confluence of the mouth of the Manacapuru River (Figure 1) in the central region of the Amazon basin. The T3 station is collocated in an area of pasture (with shallow depth roots water extraction), surrounded by native forest with about 35 m of canopy height (Martin et al. 2016)."

*e. Lines 208-211: Groundwater stress are not be the only reason for low evapotranspiration.*

Indeed, the Groundwater stress is not the only reason. Thus a new was included (P6, L211-215, text below).

"The latent heat flux (LE) showed that the majority of the available net radiation (daytime conditions) was used for this flux. Both periods had similar maximum values, with 355.9 (IOP1) and 350 W m$_{-2}$ (IOP2) at 12 LT. However, there was a reduction of net radiation converted into LE (evaporative fraction) between the periods: it was 75% in IOP1 and 66% in IOP2. Since IOP2 refers to the dry season in the region, there are a lower soil water availability, which resulted in lower LE/Rn partition in comparison IOP1."

*f. Lines 223-230: So, how mechanical turbulence will affect the NBL for some instruments, and not for others? I believe a more in-depth analysis should be performed. If there is a presence of LLJ, then more likely there will not be too much turbulent mixing, and the NBL is going to grow due to radiative cooling. Also, usually the local maxima of the LLJ is not co-located with the bottom of the residual layer, so using the level of maxima wind speed will underestimate the NBL. Last, but not least, if there is rain, then most likely the remote sensing instruments do not work properly, and they will be unable to detect any mixing height.*

Mechanical turbulence affected all instruments, however, each instrument has different measurement techniques, thus presenting differences in PBL height measurements at this range (00 - 60 LT). The study in question evaluated different PBLH measurement instruments, so the underestimation found for the NBL of the instruments that used the maximum wind speed level to estimate the height (SODAR, WP) showed that these sensors have limitations for measuring the NBL under mechanical turbulence.

*g. The "erosion" of NBL discussion seems confusing and does not make any sense to me. Looking at figures 4 and 6, if the PBL is growing above 250m after 6am, thenit means that there is no more a stable layer. If not, then how the PBL is growing? Iguess the authors can check that looking at the SODAR RASS data: : :*

According to Stull (1988), during the early morning the PBL is shallow. Its depth increases slowly at first because of the strong nocturnal stable layer that caps the young

CBL. This first phase is sometimes referred to as the erosion of the NBL, which begins after sunrise and the complete erosion occurs when the PBL has a growth rate above 100 m h$^{-1}$.

The text in question has been changed for readers' better understanding (P7, L248-253, text below).

"The phase of the erosion of the NBL according to Stull (1988) begins after sunrise (at 06 LT in Amazon) and the complete erosion occurs when the PBL has a growth rate above 100 m h$^{-1}$. In the IOP1 (Figure 4A), the erosion phase was occurred 3 hours after sunrise. only after 08 LT has occurred an increase in PBL heights (average rate of 22.8 m h$^{-1}$). This occurs due to a lower H flux (Figure 3A), which causes the erosion of the NBL to progress slowly, and total erosion occurs only at 09 LT when average rate is 102 m h$^{-1}$."

*h. Lines 254-255, 300-302: H by itself does not contributes for the PBLH growth, or the rate of change of temperature.*

Positive sensitive heat flux (H) heats the air near the surface and by convection will rise, generating turbulent movements within the PBL which increases its depth. According to standard literature on studies of PBL (Stull, 1998; Holtslag, 1987; Foken, 2008), described below:

"*During the daytime when H>0 the atmosphere is heated from below, while for H<0 the atmosphere near the surface is cooled (nighttime). This leads to unstable (daytime) and stable (nighttime) stratifications of the PBL. The magnitude and the diurnal cycle of H has important consequences on the height and the structure of PBL. During unstable conditions the air adjacent to the surface is heated and will rise. This rise continues as long as the air is warmer then the surroundings (up to an inversion layer). This process is know as convection, which is accompanied with the production of convective turbulence in the PBL.*"(Holtslag, 1987).

"*The daily cycle is highly variable, after sunrise, the atmosphere is warmed by turbulence flux H from the ground surface, and the inversion layer formed during the night brakes up (erosion phase). The new layer is very turbulent, well mixed (mixed layer), and bounded above by an entrainment zone.*"(Foken, 2008; Stull, 1998).

*i. Lines 255-257: Not sure about that a small H will decrease the PBLH. According to the theory, if there is negative flux-divergence, the PBLH will increase. Also, how the entrainment at the top was calculated?*

As H is reduced at late afternoon, then the thermal convection is not so active anymore and the PBLH starts to collapse slowly. The entrainment rate was not calculated but is is out of the goal of this analysis.

*j. Lines 310-312: If one connects the red dots in figures 4,6, he/she can see a diurnal pattern in the RS PBLH time series.*

I agree, there is a daytime cycle pattern of the PBL.

**Some Minor Concerns:**

*Lines 81-82: Wind Profiler, ceilometer, SODAR, MWP, and Lidar are not remote sensing estimation methods, but the instruments that probe the lower troposphere.*

Thanks for the observation, it was changed in the text

*Table 1: Ceilometer vertical resolution is missing. I believe some of the instrumentation measure more than reported in row 3.*

This info was included in the Table 1

*Figure 1: What is the location of the site? There are 2 red squares and red dots, are they the instrument location of Manacapuru site? If so, where are the location of the instrumentation? If now, what are they?*

Figure 1 has been modified to better present the location of the site.

*Figures 3 and 5: What are the error bars for the measurements?*

The figures has been modified to improve comprehension.

*Figures 4 and 6: What is the shaded area? The errors? If so, how the errors are estimated? Why RS points do not have error bars?*

The shaded area is errors and they have been estimated by the standard deviation of the measurements All errors/uncertainties were included in the Figure and Table X

**SC1**

*Comments on the ceilometer part:*

*Sections 2.4 and 2.6: as far as the same methodology is used (exploitation of a backscatter signal) the explanations can be combined: a ceilometer is a backscatter lidar, i.e., they follow the same physical principles.*

I agree with your observation, but one of the objectives of the paper was to compare different PBLH estimation techniques / instruments and we chose to leave the sections of the instruments separate. However, the sections have been reorganized (LIDAR - Section 2.4 and Ceilometer - Section 2.5).

*line 120: what is meant with "high frequency instrument"? As this could be misleading I suggest to write that the "temporal resolution is high" or something similar.*

It's ok. The text has been changed (L 144).

*lines 121ff: "powerful tool ... high level of detail": This is quite a general statement that neglects the problems of retrieving the MLH, in particular when a low power ceilometer (CL31 compared to CL51 or Lufft ceilometers) is used. A brief overview of the inherent pitfalls should be given: Signal artefacts (Kotthaus et al.,2016, AMT), overlap problems [relevant in particular for the NBL] and water vapor absorption (Wiegner et al., 2019, AMT), or wrong attribution of detected layers (Geiß et al., 2017, AMT).*

The description of the instrument was more detailed in the text due to yours contributions and the indicated articles.

*Section 2.4: The description how the MLH is determined from the CL31 signals is missing. Is the proprietary software BL-VIEW used? How is it applied (compare Geiß et al., 2017)? A brief outline is strongly recommended as it help to understand the accuracy of the retrieval.*

Your suggestion has been added to the text. (P4, L151-152, text below).

"The standard procedure for the PBL heights determination from Vaisala ceilometers is the software package BL-VIEW developed by the manufacturer (see more details in Morris, 2016; Geiß et al. 2017)."

*line 123: What is meant by "reflexive properties"? Do the authors mean the refractive index? It could indeed be dependent on the relative humidity but I doubt that this effect has a significant influence on the MLH-retrieval. Or is the relative humidity mentioned because of potential water vapor absorption? Again, it is unlikely that this effect is relevant for the MLH-retrieval (Wiegner et al., 2014,AMT).*

*line 124: "creating a tridimensional map": What does "tridimensional" mean? Iassume that the ceilometer provides MLH as a function of time, or the particlebackscatter coefficient as a function of time and height. Moreover, mentioning"aerosols, air pollutants, and industrial and natural emissions" might be misleading if it is interpreted as the potential to discriminate between different types ofaerosol particles; this is impossible by a simple single-wavelength backscatterlidar (ceilometer).*

*line 125: what is a "retro diffusion" laser? Just skip this word. The expressions "coefficient of the attenuated portion" and "coefficients for aerosols" arenot clear/known: do you mean "attenuated backscatter" or "particle backscatter coefficient"?*

*line 126: "subsequently the heights of ... the PBL are calculated". See my previous comment. Please outline how this has been done.*

We agree that with the changes suggested by SC-1 regarding lines 123 to 126 were necessary, and to improve the reader's understanding, they have been restructured in paragraph (P4, L144-152, text below).

"The ceilometer is a high temporal resolution instrument with a measurement interval of 2 s, and a sampling rate of 16 s and is a powerful tool for measuring the height of the PBL during its daily cycle (day and night phases) to a high level of detail. The ceilometer signal is resulting over backscattering light by particles at atmosphere, then intensity of backscattering depends on the concentration of particles in the air (Morris, 2016). Ceilometers use a retrodiffusion laser to determine the attenuated backscatter, and the particle backscatter coefficient are obtained from these data, and subsequently

the heights of the cloud base and the PBL are calculated (Wiegner et. al., 2014; Kotthaus et al., 2016; Geiß et al. 2017).

The standard procedure for the PBL heights determination from Vaisala ceilometers is the software package BL-VIEW developed by the manufacturer (see more details in Morris, 2016; Geiß et al. 2017)."

*line 145: Similar to the ceilometer-section: What is a "retro-diffused signal"?*
Text has been improved as suggested by SC1 (P4, L122-126, text below).

"These instruments employ an laser transmitter operating at a wavelength of 1.5 µm, low pulse energy (~100 µJ), and high pulse repetition frequency (15 kHz). These instruments have full upper hemispherical scanning capability and provide range-resolved measurements of attenuated particle backscatter coefficient and radial velocity. The fundamentals of its operation are similar to those of radar in which pulses of energy are transmitted to the atmosphere, the energy that is bounced back to the receiver is collected and measured as a resolved signal in time (Newsom, 2012)."

*Section 2.6: The authors mention "attenuated backscattering" (this means they use the lidar as an "elastic backscatter lidar") but do not use this quantity for the MLH-retrieval (why?)? Do I understand this correctly?*
Since the objective of this work is to estimate the height of the PBL by different methodologies for Amazonia, the calculation of the PBLH with the Lidar data was performed using the methodology of Huang et al. (2017), based on sigamaw², whose applicability was proved another location (such ..artigo! ..).

*General comments:*
*The authors should explain in detail how they determine the diurnal cycles of theMLH (for each instrument): which days are considered (only if full diurnal cyclescan be determined)? Are the sample of days the same/similar for the differentapproaches? If not, is a bias expected? Did any instrument failures occur? Whatare the reasons for the gaps of some curves?*

The description of the methodologies used to obtain the PBLH of each instrument is in the text. Every day was used, the data were verified and gap fillings was performed

Nearest-neighbor. Each IOP corresponds to 45 days of observations, as described in Table 1. The gaps in the SODAR data are due to the instrument not capturing the CBLH, and Lidar showed many gaps in the NBL.

*The authors emphasize that the precipitation is quite high at the site. When itis raining the retrieval of the MLH by a ceilometer is not possible. How do these meteorological conditions influence the number of MLH-retrievals? How many full diurnal cycles could be determined during the IOPs? In this context the "sporadic rainfall" (e.g., line 220) could be discussed in more detail.*

According of the manufacturer's manual the Ceilometer operate in different environment and weather conditions (fog, precipitation and etc.) as described to Moris, 2016 (text below).

"These instruments employ pulsed diode laser lidar (light detection and ranging) technology, where short, powerful laser pulses are sent out in a vertical or slanted direction. The directly backscattered light caused by haze, fog, mist, virga, precipitation, and clouds is measured as the laser pulses traverse the sky. This is an elastic backscatter system, and the return signal is measured at the same wavelength as the transmitted beam."

*The authors consider the MLH derived from radiosondes as truth. This is a frequently made assumption. However, the authors should explicitly mention that the methodologies (radiosonde vs. ceilometer/lidar) are based on different physical concepts.*

According to the literature, RS data are taken as reference values for PBL height studies. The other methodologies/techniques used in this study were described in the following lines:

RS (L 91-95); WP (L 104-107); SODAR (L 109-112); Deal (L 122-126); Ceilometer (L 144-150); MWR (L 154-159).

*According to the shaded areas (not explained!) in Fig. 4 the variation/uncertainty(or whatever it indicates) is so large that the rapid growth/decrease of the MLHas discussed in lines 221ff is not significant.*

The shaded represents the instrument error (changed in figures). The variation discussed in line 221 refers to the temporal variation of PBL height and not to the instrument error.

However, to improve interpretation, a table of the standard deviation of the instruments was attached, as suggested by the reviewer 1.

*MLHs of 50 − 150 m (line 234, Acevedo's results) can hardly be retrieved by aceilometer (in particular when the overlap correction function is not very accurate). So it might be possible that the NBL is at 50 m at the authors' site but notdetectable by the ceilometer. This fact is not covered by the discussion.*

The present study is not only focused on the Ceilometer, RS and the other instruments also show ceilometer equivalent heights, as well as the study by Neves and Fisch (2011) in the Amazon region of similar environmental/climatic conditions to the area of this study.

*Mentioning growth rates of e.g. 22.8 m/h (line 243) pretends an accuracy that is unrealistic. How is it determined: from the average over an IOP or from the mean of all individual diurnal cycles during the IOP (in the latter case the uncertainty can be estimated)? How is such a "precision" justified in view of the vertical resolution of the ceilometer? Please explain.*

The growth rate that I refer to in the text is calculated for each phase time interval (NBL, NBL erosion, CBL growth) of the PBL, not the daily cycle average or the IOP period average.

*The limited temporal coverage of the lidar (compared to the ceilometer) retrievals should be explained.*

Lidar's night time coverage was not displayed by many gaps.

*A comment on the availability of the different methodologies should be given: How many retrievals (obviously hourly averages) from the ceilometer, the lidar, and so on, are used for Figs. 4 and 6, (see also my first comment).*

As modified in the text in response to reviewers 1 and 2, for each instrument 45 days of observation were used for each IOP.

*Minor/technical comments:*

*Table 1: The vertical resolution of the ceilometer is given as "X". Please change.*

Changed in the Table

*Figs. 4 and 6: Explain in the caption what the dashed line means? Instead of (A) and (B) one might also use "left" and "right".*

The vertical lines represent sunrise (06 LT), sunset (18 LT) (full line) and the dashed line represent the hour on what occur erosion of nocturnal boundary layer.

*Fig. 6: panel B refers to IOP4 not IOP3.*

Changed in the Figure

---

## Referee Report (RR1)

**Observational analysis of the daily cycle of the planetary boundary layer in the central Amazon during a non-El Nino year and El Nino year (GoAmazon project 2014/5)**

Rayonil G. Carneiro, Gilberto Fisch

Second Review

Even though I do realize the authors put a strong effort to analyze the data, I do have some fundamental questions about this work. They were scattered in my previous review, and I will consolidate here (since I believe they have not been adequately addressed). My apologies for the long comments. However, I want to be thorough, since some of the responses from the 1[st] review might be due to a lack of explanation from my part.

Major Concerns:

1. The reason that I asked about the methods of the CBL/NBL height determination is because I do not understand figures 4 and 6: why the PBL is continuous? If one gets the PBL evolution in textbooks (e.g. Stull fig. 1.7), he/she will see that the CBL and NBL will start from the ground. So, there will be a discontinuity at 6am (formation of the CBL at surface), and 6pm (formation of NBL at surface). This continuous PBL might be due the lack of detection of the lidars (as someone noted the overlap problem), but it should not be a problem for the MWR, and RASS scalar profiles. The authors also might check the temperature and moisture profiles at the flux tower and see when the NBL starts. When the heat flux becomes negative at about 5pm (figs 3 and 5), the NBL will start to set at the ground (the temperature inversion at the ground), and the surface will be disconnected with the layer above. So, it is very hard to understand how the NBL will have a 300-500m inversion at 6pm. Unless, the NBL is generated by mechanical turbulence. However, this goes against the appearance of the LLJ, in which the stable layer is formed by radiative cooling (Greco et al, 1992, Boundary Layer Meteorology journal).

2. I do not understand the authors reasoning for the collapse of the CBL during the late afternoon hours: "The CBL measured by the instruments showed a decrease in its height around 16 LT. This collapse is associated with the lower intensity of the thermal convection. Although there are a reduction in H surface fluxes by this time, it is still positive (but weak). After 17 LT the H fluxes became negative and the PBL started its stable phase (NBL)". Maybe it is a problem of using aerosol backscatter as a proxy of the CBL. If not, then as stated in the previous review, H by itself does not contributes for the PBLH growth, or the rate of change of temperature, but it is its flux divergence. Using a box model frame, a positive sensible heat flux at the ground still contributes to the PBL growth. So, the collapse of the CBL during the late afternoon would be a positive entrainment flux at the top, or a horizontal divergence at the side of the box. I have not seen such CBL collapse (this is not a slow collapse, as the authors wrote in one the responses), unless there is some precipitation and/or an air mass modification. Can the authors provide the 8pm potential temperature and mixing ratio profiles from the 8pm soundings? So, one can see the height of the residual layer (RL), and a proof of such CBL collapse.

3. Also, I have some issues with the erosion of the NBL discussion during dawn period.   I notice that Carneiro et al. (2019, C-19) has a very similar plot and analysis.  Also, the fact that the remote sensing instruments do not capture the formation of the CBL at 6-8am does not mean that the CBL is not established during this period (as written in line 334-335).  The CBL is being formed at 6-8LT because there is positive H.   This positive H will erode the NBL residual layer.  So, the PBL is defined by the CBL growth – that is not shown in figures 4 and 6.  After the stable layer is dissipated, there is no reason to discuss the erosion of the NBL. Therefore, if the CBL starts to grow at 6am for IOP 2 and IOP 4, it means that the stable layer has already been dissipated (possibly because there is not a strong temperature gradient as depicted in C-19, figure 1).
Carneiro, R. G. et al. , 2019.  Erosion of the nocturnal boundary layer in the central Amazon during the dry season. Acta Amazonica,  http://dx.doi.org/10.1590/1809-4392201804453

*Old Specific Comments (my comments in italic):*

*2. Methods*

*d. Radiosonde PBL estimation, it only shows for the procedure for the convective boundary layer (CBL), what are the criteria for the nocturnal boundary layer (NBL)?*

The Radiosonde PBL estimation showed CBL and NBL the criteria was used two profile method used for CBL and NBL by Santos and Fisch, (2007), Seidel et al., (2010) and Wang et al. (2016).

*d2. According to the authors "using the vertical profiles of θ and q the height of the PBL was calculated by identifying the vertical level 94 where there was a systematic increase in potential temperature and a sudden reduction in specific humidity".   This seems criteria for the CBL height detection, but not the NBL.  Citing Seidel et al. (2007): "The top of a surface-based inversion (SBI) [Bradley et al., 1993]. While the three methods above allow for the possibility of an unstable or neutral PBL, a surface-based T inversion is a clear indicator of a stable boundary layer, whose top can define a PBL height. If an SBI is found in a sounding, the other six methods are not evaluated, as they assume a different PBL structure."*

*3. Results, discussion, conclusion: there are some parts that needs some work and/or clarification.*

*a. There are several comparisons among several parameter (precipitation, H, LE, PBLH) for wet/dry season or El Nino/non-El Nino years. However, averages are presented without any uncertainties, so the authors cannot affirm that those averages are different.*

The averages are presented with uncertainties, as the shading in figures 3 to 6 indicates the standard error of the instruments. Significant tests were made with average/standard deviations

between wet and dry periods. Which have been described in the captions of the respective figures.

*a2. So, which tests were made?  Are those statistical tests?  If so, is there any significant levels? I believe the authors should incorporate these suggestions in the text.  Just writing to reviewers does not clarify the manuscript and its future readers.*

*c. Are the CBL or residual layer (RL) shrinking for all remote sensing PBLHs during the afternoon? I wonder about this because H is still positive till about 16:30-17:00LT.*

The CBL measured by the instruments showed a decrease in its height around 16 LT. This collapse is associated with the lower intensity of the thermal convection. Although there area reduction in H surface fluxes by this time, it is still positive (but weak). After 17 LT the H fluxes became negative and the PBL started its stable phase (NBL).

*c2.  The authors must be consistent!   According to lines 251-252" "Diurnal (convective) and nocturnal (stable) conditions were separated by vertical lines at 06 and 18 LT".  According to the authors' response, the 18LT is not the delimiter of convective conditions, and it should be 17 LT, or whenever H changes sign.*

*j. Lines 310-312: If one connects the red dots in figures 4,6, he/she can see a diurnal pattern in the RS PBLH time series.*

I agree, there is a daytime cycle pattern of the PBL.

*j2. Lines 357-359: "The RS, in spite of it being a proven high-precision method, it not captures the all daily cycle evolution of the height of the PBL, due to the long-time interval between launches" is not right, still.  If the radiosonde has an upward velocity of 5 m/s, then one can launch a radiosonde every 30 minutes, or even, every 15 minutes – and will be able to probe the whole PBL for this site.*

***Minor Concerns:***

    a.  *I am not familiar with the Vaisala RS92 SVG.  However, how the radiosonde measures the dew point temperature (line 92)?  I guess there is some confusion of what is measured and what is derived in this sentence.*

    b.  *The same can be written about ceilometer (line 147).  As a colleague wrote, the BL-View is a software that estimates the mixing layer height (MLH), and not the PBL height.  The ceilometer does not measure the MLH.*

c. *Lines 273-276:* "However, Acevedo et al. (2004), also studying in a pasture site in the Amazon (in Santarém-PA), observed lower NBL heights than those from the current study (between 50 to 150 m), and this difference occurs because of different geographic conditions (Santarém suffers the influence of river breeze) ". *However, Manaus is also influenced by the River Breeze circulation, no? According to Oliveira and Fitzjarrald (1992, Boundary Layer Meteorology journal), this circulation is responsible for the LLJ formation.*

d. *Figure captions:* "instrument error" *seems not accurate. For instance, which instrument measure turbulent fluxes? What is the error for the net radiometer? Probably much less than the Rn variation in 1 hour – principally during daytime. I believe should be "parameter uncertainties", or better, "standard deviation". No reason to let readers wondering about the meaning of "instrument error".*

e. *The standard deviations values from tables 2 and 3 seem too small for the shaded areas in figures 4 and 6, don't they?*

---

## Author Response (AR2)

Dear Mr

ACP Editor

Firstly, I would like to thank the new scientific review, which contributes to a better article for sure. Below I describe item by item, the questions of the referee. My answers are in blue and the changes in the text also are in blue color.

**Below are the responses to referee**

**Major Concerns:**

1. The reason that I asked about the methods of the CBL/NBL height determination is because I do not understand figures 4 and 6: why the PBL is continuous? If one gets the PBL evolution in textbooks (e.g. Stull fig. 1.7), he/she will see that the CBL and NBL will start from the ground. So, there will be a discontinuity at 6am (formation of the CBL at surface), and 6pm (formation of NBL at surface). This continuous PBL might be due the lack of detection of the lidars (as someone noted the overlap problem), but it should not be a problem for the MWR, and RASS scalar profiles. The authors also might check the temperature and moisture profiles at the flux tower and see when the NBL starts. When the heat flux becomes negative at about 5pm (figs 3 and 5), the NBL will start to set at the ground (the temperature inversion at the ground), and the surface will be disconnected with the layer above. So, it is very hard to understand how the NBL will have a 300-500m inversion at 6pm. Unless, the NBL is generated by mechanical turbulence. However, this goes against the appearance of the LLJ, in which the stable layer is formed by radiative cooling (Greco et al, 1992, Boundary Layer Meteorology journal).

The reviewer´s observation is correct. However, the continuous measurements (like ceilometer) of the sensors during the hours of 17 to 18 HL capture the CBL (in its decay phase forming RL) and the NBL begins to form (from the surface), as shown in Figure S1 of the supplementary document (figure below). This figure shows the measurements at 16 second intervals. As one of the objectives of the paper is to have a complete daily cycle of PBL (including CBL and NBL), we decide to leave the continuous line. However, this observation was indicated in the text.

*Remote sensors capture multi-layers from the heights of the PBL in the transition interval (day-night, between 17 and 18 LT). Figure S1 (presented at the supplementary material) shows the heights obtained through the ceilometer every 16 s, where the blue points represent the CBL (that is in its decay phase or Residual Layer) and the red points the NBL that is forming from the surface. However, as the one of the goal of this paper is to have a complete picture of the PBL cycle, the NBL heights were neglected in Figures 4 and 6, in order to show only the decay of the CBL convection.*

[Figure]

Figure S1 Height of the PBL for the sunset transition, with the blue dots referring to CBL and the red dots referring to NBL.

2. I do not understand the authors reasoning for the collapse of the CBL during the late afternoon hours: "The CBL measured by the instruments showed a decrease in its height around 16 LT. This collapse is associated with the lower intensity of the thermal convection. Although there are a reduction in H surface fluxes by this time, it is still positive (but weak). After 17 LT the H fluxes became negative and the PBL started its stable phase (NBL)". Maybe it is a problem of using aerosol backscatter as a proxy of the CBL. If not, then as stated in the previous review, H by itself does not contributes for the PBLH growth, or the rate of change of temperature, but it is its flux divergence. Using a box model frame, a positive sensible heat flux at the ground still contributes to the PBL growth. So, the collapse of the CBL during the late afternoon would be a

positive entrainment flux at the top, or a horizontal divergence at the side of the box. I have not seen such CBL collapse (this is not a slow collapse, as the authors wrote in one the responses), unless there is some precipitation and/or an air mass modification. Can the authors provide the 8pm potential temperature and mixing ratio profiles from the 8pm soundings? So, one can see the height of the residual layer (RL), and a proof of such CBL collapse.

As requested by the reviewer in order to answer the question were made the potential temperature and specific humidity profiles of the IOPS at 20 LT (Figures 2 below). Additional figures for wind u* and windspeed were also added in the range of 16 to 06 LT (Figures 3 and 4 below).

The vertical profile of potential temperature (Figure 2 A) and specific humidity (Figure 2 B) made with the radiosonde launched at 20 LT also showed this residual layer. In complement, the temporal variation of the u* and windspeed (Figures 3 and 4, respectively) showed that the collapse of the CBL is reduced by the mechanical turbulence that occurs near the sunset transition.

A)                                                                    B)

[Figure]

Figure 2 Vertical profile of potential temperature (A) specific humidity (B) at 20 local time of the 4 IOPs

[Figure]

Figure S3 Average of the u* (m/s) during transition from night to day of the 4 IOPs.

[Figure]

Figure S4 Average of the wind speed (m/s) during transition from night to day of the 4 IOPs.

3. Also, I have some issues with the erosion of the NBL discussion during dawn period. I notice that Carneiro et al. (2019, C-19) has a very similar plot and analysis. Also, the fact that the remote sensing instruments do not capture the formation of the CBL at 6-8am does not mean that the CBL is not established during this period (as written in line 334-335). The CBL is being formed at 6-8LT because there is positive H. This positive H will erode the NBL residual layer. So, the PBL is defined by the CBL growth – that is not shown in figures 4 and 6. After the stable layer is dissipated, there is no reason to discuss the erosion of the NBL. Therefore, if the CBL starts to grow at 6am for IOP 2 and IOP 4, it means that the stable layer has already been dissipated (possibly because there is not a strong temperature gradient as depicted in C-19, figure 1).

Carneiro, R. G. et al. , 2019. Erosion of the nocturnal boundary layer in the central Amazon during the dry season. Acta Amazonica, http://dx.doi.org/10.1590/1809-4392201804453

According to Stull (1988), the complete erosion of the NBL occurs when the whole layer is mixed, the potential vertical gradient is almost null and as a consequence there is a high growth rate (like 100 m $h_{-1}$). So, before (say 2-3 hours) after sunrise, there are still several layers (see Figure 1.7 and the S4 point in a Figures 1.11 from Stull, 1988, figures below). In order to be more elucidated, a figure of the PBL heights was added to the supplementary document every 10 minutes of the day-night transition period (Figure S3) and the H flux in the same interval (Figure S4) (figures below). Nevertheless, we have included a statement to clarify this point (text below).

*The transition from nighttime to daytime is very complex. Although the H has become positive (so heating the atmosphere), this amount of energy did not completely warm the atmosphere, eroding the previous (or old) NBL. So, the height of PBL was considered to be (still) NBL (see Figures S3 and S4 in the supplementary material). The PBL height will be CBL only when the NBL is completely eroded.*

[Figure]

Figure 1.7 The boundary layer height consists of three major parts: a very turbulent mixed layer; a less-turbulent residual layer containing former mixed-layer air; and a nocturnal stable boundary layer of sporadic turbulence.

[Figure]

Figure 1.12 Profiles of mean virtual potential temperature $\theta v$, showing the boundary-layer evolution during a diurnal cycle starting at about 16 local time identify each sounding with an associated launch time indicated in Figure 1.7.

[Figure]

Figure S3 Transition from night to day PBL of the ceilometer for 4 IOPs, averaging every 10 minutes.

[Figure]

Figure S4 Average of the sensible heat flux (H) (W m-2) during transition from night to day of the 4 IOPs.

***Old Specific Comments (my comments in italic):***

*2. Methods*

*d. Radiosonde PBL estimation, it only shows for the procedure for the convective boundary layer (CBL), what are the criteria for the nocturnal boundary layer (NBL)?*

The Radiosonde PBL estimation showed CBL and NBL the criteria was used two profile method used for CBL and NBL by Santos and Fisch, (2007), Seidel et al., (2010) and Wang et al. (2016).

*d2. According to the authors "using the vertical profiles of θ and q the height of the PBL was calculated by identifying the vertical level 94 where there was a systematic increase in potential temperature and a sudden reduction in specific humidity". This seems criteria for the CBL height detection, but not the NBL. Citing Seidel et al. (2007): "The top of a surface-based inversion (SBI) [Bradley et al., 1993]. While the three methods above allow for the possibility of an unstable or neutral PBL, a surface-based T inversion is a clear indicator of a stable boundary layer, whose top can define a PBL height. If an SBI is found in a sounding, the other six methods are not evaluated, as they assume a different PBL structure."*

Good point. I agree that the profile method is only applied for CBL. And, the text did not make it clear to readers that the method used for NBL was different. Therefore, this part of the text has been rewritten, detailing the methods for the two phases of the PBL (text below).

*From these data, the potential temperature (θ) and specific humidity (q) were extrapolated and their vertical profiles were used for the determination of e height of the PBL. At the CBL phase, the heights were identified by the vertical level where there was a systematic increase in potential temperature and a sudden reduction in specific humidity, and this method is called the profile method, as described in detail by Santos and Fisch, (2007), Seidel et al., (2010) and Wang et al. (2016). However, at the NBL phase, the heights were determined by the height where the vertical θ gradient was null or less than a defined number (0.01 K m$_{-1}$) starting from the surface. This statement relates the maximum distance from the surface where the radioactive night cooling operates, as described in detail by Santos and Fisch (2007), Neves and Fisch (2011).*

*3. Results, discussion, conclusion: there are some parts that needs some work and/or clarification.*

*a. There are several comparisons among several parameter (precipitation, H, LE, PBLH) for wet/dry season or El Nino/non-El Nino years. However, averages are presented without any uncertainties, so the authors cannot affirm that those averages are different.*

The averages are presented with uncertainties, as the shading in figures 3 to 6 indicates the standard error of the instruments. Significant tests were made with average/standard deviations between wet and dry periods. Which have been described in the captions of the respective figures.

*a2. So, which tests were made? Are those statistical tests? If so, is there any significant levels? I believe the authors should incorporate these suggestions in the text. Just writing to reviewers does not clarify the manuscript and its future readers.*

We performed a statistical test (test t), comparing the sensors data with the RS (defined as true data). In this way, following the suggestion given by the reviewer, it was incorporated into the text (text below) and all test tables are in the supplementary document (Table S1 to S4).

*In the results obtained, the average and standard deviations values were computed for different time intervals along the PBL daily cycle (Tables 2 and 3). Also, a significant test (Test t-Student) was applied to assess the significance of the correlation coefficients amongst the remote sensors compared to the RS (Tables S1 to S4 in the supplementary material).*

*c. Are the CBL or residual layer (RL) shrinking for all remote sensing PBLHs during the afternoon? I wonder about this because H is still positive till about 16:30-17:00LT.*

The CBL measured by the instruments showed a decrease in its height around 16 LT. This collapse is associated with the lower intensity of the thermal convection. Although there area reduction in H surface fluxes by this time, it is still positive (but weak). After 17 LT the H fluxes became negative and the PBL started its stable phase (NBL).

*c2. The authors must be consistent! According to lines 251-252" "Diurnal (convective) and nocturnal (stable) conditions were separated by vertical lines at 06 and 18 LT". According to the authors' response, the 18LT is not the delimiter of convective conditions, and it should be 17 LT, or whenever H changes sign.*

I thank you for the observation because the text was confusing. The vertical lines were chosen to delimit the time of sunrise and sunset, which does not vary much between the

periods as the site is near the Equator. Hence, the text was corrected to make it consistent for readers (text below).

*In which, the sunrise and sunset times were marked by the vertical lines of 06 and 18 LT respectively, since the study area is near to the equator line and there are not changes at these times.*

*j. Lines 310-312: If one connects the red dots in figures 4,6, he/she can see a diurnal pattern in the RS PBLH time series.*

I agree, there is a daytime cycle pattern of the PBL.

*j2. Lines 357-359: "The RS, in spite of it being a proven high-precision method, it not captures the all daily cycle evolution of the height of the PBL, due to the long-time interval between launches" is not right, still. If the radiosonde has an upward velocity of 5 m/s, then one can launch a radiosonde every 30 minutes, or even, every 15 minutes – and will be able to probe the whole PBL for this site.*

I agree that with launches every 30 minutes of RS it would be possible to probe the entire PBL cycle. However, GoAmazon launched only during synoptic times, and it is not possible to obtain the daily cycle through RS. We are focusing on the high temporal resolution (say 10-15 min) from the remote sensors instruments. Even so, the text was changed to show the limitation due to the number of launches per day (text below).

*The RS, in spite of it being a proven high-precision method, in this experiment it was launched only on synoptic times plus an extra at 15 UTC. Hence, it did not capture a high temporal resolution (like the remote sensors) daily cycle evolution of the height of the PBL, due to the long time interval between launches (each 6 h).*

***Minor Concerns:***

*a. I am not familiar with the Vaisala RS92 SVG. However, how the radiosonde measures the dew point temperature (line 92)? I guess there is some confusion of what is measured and what is derived in this sentence.*

Good point. The text really confuses the readers. The dew point temperature is a derived variable from the humidity sensor. Thus, the text has been corrected to improve the reader's understanding (text below).

*From the RS measurements, the following data were measured as functions of time during a free-balloon ascent: pressure (hPa), air temperature (dry bulb) (ºC), relative humidity (%), wind velocity (m s−1) and wind direction (deg). With these measurements, others derived quantities were computed and used in this study: altitude (m), geographic position (latitude and longitude), dew point temperature (ºC), u-component of wind velocity (m s−1) and v-component of wind velocity (m s−1).*

*b. The same can be written about ceilometer (line 147). As a colleague wrote, the BL-Viewis a software that estimates the mixing layer height (MLH), and not the PBL height. The ceilometer does not measure the MLH.*

According to the Ceilometer´s Handbook (Morris, 2016) used in the GOAmazon experiment, the height of the PBL is estimated through the optical backscattering of the instrument. As the MLH is the major part of the diurnal PBL, it may be a misunderstanding issue. Thus the text has been modified to improve comprehension (see the new text below).

*These measurements are used to produce derived products that are recorded: the height of the cloud base, the retrieval of the particle backscatter coefficient and PBL height. Although the ceilometer measures the reflection of the aerosols layer (so the mixing layer height), it was assumed as the diurnal PBL height) as the entrainment zone is very shallow. So, during all manuscript, this information (backscatter aerosols) was assumed as PBL height.*

*c. Lines 273-276:* "However, Acevedo et al. (2004), also studying in a pasture site in the Amazon (in Santarém-PA), observed lower NBL heights than those from the current study (between 50 to 150 m), and this difference occurs because of different geographic conditions (Santarém suffers the influence of river breeze) ". *However, Manaus is also influenced by the River Breeze circulation, no? According to Oliveira and Fitzjarrald (1992, Boundary Layer Meteorology journal), this circulation is responsible for the LLJ formation.*

I agree with your observation. Manacapuru (T3 site during GOAmazon experiment) can also be influenced by the river breeze, but not as strong as Manaus or Santarém. So the text was changed.

*However, Acevedo et al. (2004), also studying in a pasture site in the Amazon (in Santarém-PA), observed lower NBL heights than those from the current study (between 50 to 150 m), and this difference occurs because of different geographic conditions (influence of river breeze, fog formation, etc). An example of these influences, in the Santarém region the authors related several cases of formation of fog during the night, which was not observed at the pasture site in Rondônia (see Neves and Fisch, 2015) nor at T3 site.*

*d. Figure captions: "instrument error" seems not accurate. For instance, which instrument measure turbulent fluxes? What is the error for the net radiometer? Probably much less than the Rn variation in 1 hour – principally during daytime. I believe should be "parameter uncertainties", or better, "standard deviation". No reason to let readers wondering about the meaning of "instrument error".*

Good point. The "instrument error" can leave the reader with doubts, so following your suggestion I changed the terms to "**standard deviation**" in all manuscript.

*e. The standard deviations values from tables 2 and 3 seem too small for the shaded areas in figures 4 and 6, don't they?*

Your observation is correct. The values have been revised in the text. But tables 2, 3 showed the average of the standard deviations for different intervals from PBL daily cycle, whereas the shaded areas in Figure 4 and 6 are related on average every 30 min.

*The shaded area for Figure 3 which represented the standard deviations values, were computed for each 30 min time interval. It is also shown in the following Figures 4 to 6.*

---

## Author Response (AR3)

Dear Mrs. ACP Editor

Firstly, we would like to thank the new scientific review, which contributes to a better article for sure. Sorry, we did not address properly the problem of the statistical test during the last review. We apologize for that. Below the modifications were described and the answers are in blue and the changes in the text also are in blue color.

Editor Decision:

One point that still requires some clarification is regarding the text introduce in lines **200-201: "Also, a significant test (Test t-Student) was applied to assess the significance of the correlation coefficients amongst the remote sensors compared to the RS (Tables S1 to S4 in the Supplement)."** The tables S1-S4 do not have a complete label that allows the reader to understand the numbers and does not indicate what numbers are significant. Please provide a complete table caption, with p-value, degrees of freedom, etc. Also, the manuscript does not provide any comment on the results in Table S1-S4, regarding the significance of the correlations.

Good point, we thanks for this observation. The Tables are not self-explanatory indeed. Thus, the legends of the Tables S1 to S4 (below) have been modified in the supplement document to improve the reader's understanding. As requested, the p-value and the degrees of freedom were added. Also, a new comment on these results was also added in the manuscript (Text below).

*In the results obtained, the average and standard deviations values were computed for different time intervals along the PBL daily cycle (Tables 2 and 3). The computed Pearson's correlation coefficient (r) showed values higher than 0.6 for all remote sensors related to the RS, especially for the ceilometer which showed correlations around 0.8. Also, a significant statistical test (t-Student with 95%) was applied for the 45 days of each IOP, with 2 degrees of freedom, and the results showed that there is statistical significance between the remote sensors and RS (Tables S1 to S4 in the Suplement).*

**Tabela S1 Student t-test statistics calculated to all different instruments of PBL height in comparison to RS during IOP1 with 2 degrees of freedom. Pearson's correlation coefficient is represented by $r$, $t_c$ is the critical value and p-value is the probability value.**

| | IOP1 (February 15 to March 31, 2014) | | | | | | | | | | | | | | |
|---|---|---|---|---|---|---|---|---|---|---|---|---|---|---|---|
| **Hours** | **02 LT** | | | **08 LT** | | | **11 LT** | | | **14 LT** | | | **20 LT** | | |
| **Sensors** | $r$ | $\pm t_c$ | p-value* | $r$ | $\pm t_c$ | p-value* | $r$ | $\pm t_c$ | p-value* | $r$ | $\pm t_c$ | p-value* | $r$ | $\pm t_c$ | p-value* |
| Ceilometer | 0.95 | 1.679 | 0.27 | 0.90 | 1.301 | 0.54 | 0.97 | 2.014 | 0.75 | 0.97 | 2.014 | 0.12 | 0.95 | 1.679 | 0.19 |
| Lidar | -X- | -X- | -X- | 0.95 | 1.679 | 0.68 | 0.97 | 2.014 | 0.70 | 0.95 | 1.679 | 0.35 | -X- | -X- | -X- |
| MWR | -X- | -X- | -X- | -X- | -X- | -X- | -X- | -X- | -X- | -X- | -X- | -X- | -X- | -X- | -X- |
| RWP | 0.90 | 1.301 | 0.44 | 0.80 | 0.850 | 0.13 | 0.90 | 1.301 | 0.22 | 0.95 | 1.679 | 0.47 | 0.90 | 1.301 | 0.37 |
| Sodar | 0.75 | 0.680 | 0.23 | -X- | -X- | -X- | -X- | -X- | -X- | -X- | -X- | -X- | 0.80 | 0.850 | 0.17 |

\* Confidence interval considered: 95% ($\propto$ = 0.05). In order to have statistical significance, the tests of null hypothesis must have p-value > $\propto$ = 0.05.

"X" represents where absences measurements occurred.

**Tabela S2 Student t-test statistics calculated to all different instruments of PBL height in comparison to RS during IOP2 with 2 degrees of freedom. Pearson's correlation coefficient is represented by $r$, $t_c$ is the critical value and p-value is the probability value.**

| | IOP2 (September 1 to October 15, 2014) | | | | | | | | | | | | | | |
|---|---|---|---|---|---|---|---|---|---|---|---|---|---|---|---|
| **Hours** | **02 LT** | | | **08 LT** | | | **11 LT** | | | **14 LT** | | | **20 LT** | | |
| **Sensors** | $r$ | $\pm t_c$ | p-value* | $r$ | $\pm t_c$ | p-value* | $r$ | $\pm t_c$ | p-value* | $r$ | $\pm t_c$ | p-value* | $r$ | $\pm t_c$ | p-value* |
| Ceilometer | 0.95 | 1.679 | 0.25 | 0.97 | 2.014 | 0.53 | 0.99 | 2.412 | 0.90 | 0.99 | 2.412 | 0.85 | 0.90 | 1.301 | 0.20 |
| Lidar | -X- | -X- | -X- | 0.95 | 1.679 | 0.49 | 0.90 | 1.301 | 0.60 | 0.95 | 1.679 | 0.47 | -X- | -X- | -X- |
| MWR | 0.95 | 1.679 | 0.67 | 0.95 | 1.679 | 0.45 | 0.90 | 1.301 | 0.25 | 0.90 | 1.301 | 0.38 | 0.95 | 1.679 | 0.32 |
| RWP | 0.80 | 0.850 | 0.22 | 0.95 | 1.679 | 0.46 | 0.90 | 1.301 | 0.41 | 0.90 | 1.301 | 0.32 | 0.75 | 0.680 | 0.12 |
| Sodar | 0.80 | 0.850 | 0.17 | -X- | -X- | -X- | -X- | -X- | -X- | -X- | -X- | -X- | 0.90 | 1.301 | 0.56 |

\* Confidence interval considered: 95% ($\propto$ = 0.05). In order to have statistical significance, the tests of null hypothesis must have p-value > $\propto$ = 0.05.

"X" represents where absences measurements occurred.

**Tabela S3 Student t-test statistics calculated to all different instruments of PBL height in comparison to RS during IOP3 with 2 degrees of freedom. Pearson's correlation coefficient is represented by $r$, $t_c$ is the critical value and p-value is the probability value.**

IOP3 (February 15 to March 31, 2014)

| Hours | 02 LT | | | 08 LT | | | 11 LT | | | 14 LT | | | 20 LT | | |
|---|---|---|---|---|---|---|---|---|---|---|---|---|---|---|---|
| Sensors | $r$ | $\pm t_c$ | p-value* | $r$ | $\pm t_c$ | p-value* | $r$ | $\pm t_c$ | p-value* | $r$ | $\pm t_c$ | p-value* | $r$ | $\pm t_c$ | p-value* |
| Ceilometer | 0.97 | 2.014 | 0.62 | 0.80 | 0.850 | 0.88 | -X- | -X- | -X- | 0.95 | 1.679 | 0.47 | 0.97 | 2.014 | 0.54 |
| Lidar | -X- | -X- | -X- | 0.95 | 1.679 | 0.35 | -X- | -X- | -X- | 0.90 | 1.301 | 0.47 | -X- | -X- | -X- |
| MWR | 0.80 | 0.850 | 0.39 | 0.90 | 1.301 | 0.42 | -X- | -X- | -X- | 0.80 | 0.850 | 0.36 | 0.70 | 0.528 | 0.33 |
| RWP | 0.80 | 0.850 | 0.25 | 0.90 | 1.301 | 0.33 | -X- | -X- | -X- | 0.80 | 0.850 | 0.27 | 0.90 | 1.301 | 0.31 |
| Sodar | 0.70 | 0.528 | 0.10 | -X- | -X- | -X- | -X- | -X- | -X- | -X- | -X- | -X- | 0.70 | 0.528 | 0.11 |

* Confidence interval considered: 95% ($\propto$ = 0.05). In order to have statistical significance, the tests of null hypothesis must have p-value > $\propto$ = 0.05.

"X" represents where absences measurements occurred.

**Tabela S4 Student t-test statistics calculated to all different instruments of PBL height in comparison to RS during IOP4 with 2 degrees of freedom. Pearson's correlation coefficient is represented by $r$, $t_c$ is the critical value and p-value is the probability value.**

IOP4 (September 1 to October 15, 2015)

| Hours | 02 LT | | | 08 LT | | | 11 LT | | | 14 LT | | | 20 LT | | |
|---|---|---|---|---|---|---|---|---|---|---|---|---|---|---|---|
| Sensors | $r$ | $\pm t_c$ | p-value* | $r$ | $\pm t_c$ | p-value* | $r$ | $\pm t_c$ | p-value* | $r$ | $\pm t_c$ | p-value* | $r$ | $\pm t_c$ | p-value* |
| Ceilometer | 0.97 | 2.014 | 0.35 | 0.97 | 2.014 | 0.61 | -X- | -X- | -X- | 0.99 | 2.412 | 0.82 | 0.99 | 2.412 | 0.27 |
| Lidar | -X- | -X- | -X- | -X- | -X- | -X- | -X- | -X- | -X- | -X- | -X- | -X- | -X- | -X- | -X- |
| MWR | 0.90 | 1.301 | 0.59 | 0.80 | 0.850 | 0.50 | -X- | -X- | -X- | 0.80 | 0.850 | 0.42 | 0.99 | 2.412 | 0.39 |
| RWP | 0.80 | 0.850 | 0.44 | 0.90 | 1.301 | 0.54 | -X- | -X- | -X- | 0.70 | 0.528 | 0.32 | 0.97 | 2.014 | 0.38 |
| Sodar | 0.80 | 0.850 | 0.21 | -X- | -X- | -X- | -X- | -X- | -X- | -X- | -X- | -X- | 0.80 | 0.850 | 0.17 |

* Confidence interval considered: 95% ($\propto$ = 0.05). In order to have statistical significance, the tests of null hypothesis must have p-value > $\propto$ = 0.05.

"X" represents where absences measurements occurred.